# Unified Source-Free Domain Adaptation via Causality-Inspired Latent Invariant Factor Learning

## Abstract

In the pursuit of transferring a source model to a target domain without access to the source training data, Source-Free Domain Adaptation (SFDA) has been extensively explored across various scenarios, including Closed-set, Open-set, Partial-set, Open-partial-set, and Generalized settings. Existing methods, focusing on specific scenarios, not only address a limited subset of challenges but also necessitate prior knowledge of the target domain, significantly limiting their practical utility and deployability. In light of these considerations, we introduce a more practical yet challenging problem, termed *unified SFDA*, which comprehensively incorporates all specific scenarios in a unified manner. In this paper, we propose a novel approach *Causality-inspired latent invariant factor learning for unified SFDA* (CausalDA). In contrast to previous alternatives that emphasize learning the statistical description of reality, we formulate CausalDA from a causality perspective. The objective is to uncover potential causality-induced connections between latent invariant variables and model decisions, enhancing the reliability and robustness of the learned model against domain shifts. To integrate extensive world knowledge, we leverage a pre-trained vision-language model such as CLIP. This aids in the formation and discovery of latent invariant factors in the absence of supervision in the variation of distribution and semantics, coupled with a newly designed self-supervised invariant information maximization with theoretical guarantees. Extensive experiments demonstrate that CausalDA can achieve new state-of-the-art results in distinct SFDA settings, as well as source-free out-of-distribution generalization.

## 1 Introduction

Due to demands for privacy and information protection in public security and commercial competition, well-annotated training data are recognized as crucial for varying purposes, requiring stringent access control. In practical scenarios, the use of a source model (pre-trained on the source domain) without access to the actual source data, referred to as Source-Free Domain Adaptation (SFDA) (Kim et al., 2021), has become more feasible. This approach involves the challenges posed by strict data access controls and emphasizes adapting a pre-trained model to new domains without relying on the availability of the original source data.

SFDA has been explored across various scenarios, as summarized in Tab. 1. The most constrained scenario, termed *Closed-set*, assumes identical categories of interest in both the source and target domains, aiming to address covariate shift, such as domain-specific appearance distributions (Li et al., 2020b). *Generalized SFDA* extends this by requiring the adapted model to perform well in both the source and target domains without forgetting the source domain (Yang et al., 2021b). Moving beyond the vanilla category identity con-

Table 1: Summary of different SFDA settings. $C_s/C_t$: The category set of the source/target domain, $C = C_s \cap C_t$; AF is short for Anti-Forgetting; Covariate-shift and Semantic-shift refer to the distinction in data probability distribution and category composition, respectively.

| Setting | Covariate-shift | Semantic-shift | AF | Category |
|---|---|---|---|---|
| Closed-set | ✓ | ✗ | ✗ | $C_s = C_t$ |
| Generalized | ✓ | ✗ | ✓ | $C_s = C_t$ |
| Open-set | ✓ | ✓ | ✗ | $C_s \subset C_t$ |
| Partial-set | ✓ | ✓ | ✗ | $C_s \supset C_t$ |
| Open-partial-set | ✓ | ✓ | ✗ | $|C|/|C_s \cup C_t| \in (0,1)$ |
| **Unified (Ours)** | ✓ | ✓ | ✓ | Any |

straint, *Open-set* SFDA (Panareda Busto & Gall, 2017) considers the target domain with additional categories, while *Partial-set* SFDA (Cao et al., 2019) takes the opposite direction. Both scenarios require handling the additional challenge of semantic shift. Synthesizing the two orthogonal dimensions leads to *Open-Partial-Set* (Universal) SFDA (Kundu et al., 2020), a generalized setting that explicitly accounts for unknown category relationships.

Typically, prevalent SFDA methods tend to concentrate on particular scenarios, tackling only a subset of challenges. For instance, SFDA-DE(Ding et al., 2022) focuses on Closed-set scenarios, SF-PGL(Luo et al., 2023) on Open-set scenarios, PSAT(Tang et al., 2024b) on Generalized settings, CRS (Zhang et al., 2023) on Partial-set scenarios, and LEAD (Qu et al., 2024) on Open-partial-set scenarios. Nevertheless, this approach significantly restricts their practical utility and deployability, given that we often possess limited prior knowledge about the target domain and minimal control or selection over its conditions.

To address the aforementioned limitation, this study introduces a more realistic yet challenging problem, referred to as *Unified Source-Free Domain Adaptation* (*Unified SFDA*), with the goal of comprehensively addressing all the specific scenarios mentioned in a unified manner. To achieve this, we propose a novel approach *latent Causal discovery of salient factors for unified SFDA (CausalDA)*, going beyond conventional statistical association learning of related variables by exploring the underlying causal relationships (Pearl, 2009). The causal mechanism discovered is expected to be more reliable in varying distributions and semantic contexts, providing a unified solution for different scenarios of SFDA.

In the absence of label supervision for both distributional and semantic variations, our CausalDA is specifically formulated from a structural causal hypothesis in the logits space. Crucially, rather than attempting direct causal recovery, we aim to learn latent invariant factors that serve as operational proxies for the underlying causal variables. Considering that the information from both domains is not necessarily complete, these proxy factors are disentangled into two complementary parts: (i) *external invariant factors* and (ii) *internal invariant factors*. For the former, we leverage a pre-trained large Vision-Language (ViL) model with rich knowledge such as CLIP (Radford et al., 2021), which has been exposed to a vast amount of multimodality information sources. The latter is identified under the guidance of the learned external proxies. The estimation of these external and internal invariant factors is alternated by designing a self-supervised invariant information maximization with theoretical guarantees.

Our **contributions** are summarized as follows:

(i) We introduce a unified SFDA problem that aligns with practical demands of real-world deployment, ranging from focus on unknown category relationships to source-knowledge preservation and out-of-distribution robustness. By eliminating the need for target-domain priors, our design significantly enhances deployability and versatility in unpredictable real-world environments.

(ii) We propose a novel CausalDA approach for unified SFDA. Instead of learning the statistical description of problem reality as conventional methods do, CausalDA is formulated under a causality perspective. It attempts to reveal the potential causality-induced relationship between latent variables and model decisions, providing favorable robustness against both distributional and semantic shifts.

(iii) Extensive experiments demonstrate that CausalDA consistently achieves new state-of-the-art results on varying SFDA scenarios as well as source-free out-of-distribution generalization.

## 2 Related Work

### 2.1 Source-free domain adaptation

Most of the prior SFDA methods consider the Closed-set setting where the source and target domains share the same classes, and the focus is on cross-domain distribution alignment. Existing methods exist in two categories. The first converts SFDA to Unsupervised Domain Adaptation (UDA) by constructing a pseudo-source domain (Li et al., 2020a), exploiting source prototype-guided data generation (Tian et al., 2021) or splitting source-like subset from target data (Du et al., 2024). The second follows self-supervised learning (Liang et al., 2020; Chen et al., 2022b), introducing self-guidance. Besides widely used pseudo-

labels (Liang et al., 2020; Litrico et al., 2023), geometry information (Yang et al., 2021a; 2023; Tang et al., 2022), contrastive data (Zhang et al., 2023) and historical data (Tang et al., 2024b) have also been exploited.

Recently, more practical settings have been studied. For example, the Generalized setting shifts aim to mitigate forgetting the source domain (Yang et al., 2021b). A representative approach is combining continual learning techniques with cross-domain adaptation, e.g., the domain attention-based gradient regulation (Yang et al., 2021b) and source guidance constructed by historical information refining (Tang et al., 2024b). There exist works considering Open-set (Luo et al., 2023), Partial-set (Li & Chen, 2023) and Open-partial-set (Kundu et al., 2020) settings. For instance, to control the Open-set risk, a progressive balanced pseudo-labeling strategy (Luo et al., 2023), and Closed-set class prototypes are proposed (Vray et al., 2024). Estimating the target data distribution helps with negative transfer reduction from the partial classes shift (Lee et al., 2022). Feature decomposition adaptively identifies the target-private data (Qu et al., 2024).

Alternatively, some works were proposed to enhance the discriminating ability for out-of-distribution classes (Samadh et al., 2023). There are two main strategies: Expanding inter-class distance to prevent semantic confusion (Tang et al., 2021; Liang et al., 2020), and introducing external semantics, e.g., CLIP, to match out-of-distribution (Shu et al., 2022).

Most recently, multimodal large models such as CLIP and ShareGPT (Chen et al., 2024b) have been introduced to advance SFDA, substantially enhancing adaptation performance by leveraging generic knowledge in these models. Two strategies are adopted: (1) customizing and adapting this knowledge via mutual knowledge distillation (Tang et al., 2024c; Zhang et al., 2025) or multi-teacher guidance (Chen et al., 2024a) or multimodal space alignment (Chen et al., 2026); and (2) further denoising ViL predictions (Tang et al., 2025). Critically, we highlight a couple of key aspects that make CausalDA conceptually distinct from these SFDA methods. First, CausalDA is unique in design philosophy by learning a causal representation, in contrast to the statistical knowledge DIFO (Tang et al., 2024c) and ProDe (Tang et al., 2025) both are designed to learn. As a consequence, the outreach and generalization can be further extended, leading to the first dedicated unified SFDA approach. Instead, DIFO and ProDe predominantly focus on the conventional closed-set setting. Further, this conceptual discrepancy leads to distinct leveraging of ViL knowledge. DIFO and ProDe both employ the generic ViL knowledge directly as supervision, fostering the establishment of statistical relationships, whilst CausalDA utilizes ViL knowledge to uncover the underlying causal factors for capturing the essence of visual recognition.

Existing work targets only specific SFDA settings, limiting its practical applicability and generality. This paper introduce a unified SFDA setting that incorporates all the pre-existing scenarios. With this new setup, we aim to foster more advanced SFDA methods that could generalize to a variety of problem settings without the need for designing and maintaining an array of distinct algorithms.

### 2.2   Causality methods in transfer learning

Causality methods aim for a robust function relation between random variables, which is insensitive to the extra variation/disturbing (Schölkopf et al., 2021). This property leads to recent popularity of this new paradigm in transfer learning, e.g., UDA (Wu et al., 2024), domain generalization (DG) (Christiansen et al., 2021), and out-of-distribution (OOD) (Byun et al., 2025). These attempts aim to reduce non-causal factors by introducing artificial intervention. For example, the probability distribution change (used to represent the domain shift) and the semantic consistency constraint are jointly employed to implement this intervention. As source labels are available to ensure consistency, these methods emphasize constructing the distribution change in two lines. One explicitly generates augmented data to disturb the distribution, e.g., the non-linear augmentation and spatially-variable blending augmentation (Ouyang et al., 2022), inverse Fourier transformation augmentation (Lv et al., 2022), generative models-based cross-domain image style transformation (Wang et al., 2022a), and causality-adjusted augmentation (Zhu et al., 2025). The other implicitly modifies the distribution by exploiting cross-domain data. For example, MatchDG (Mahajan et al., 2021) developed cross-domain contrastive learning where the input's positive sample was randomly selected from samples with the same class. COR (Wang et al., 2022b) adopted the variational inference encoder to infer unobserved causal factors from historical data, taking the time domain as a natural distribution change.

Compared with the existing works, the following three features distinguish CausalDA from them. (i) CausalDA does not rely on real labels that are indispensable for previous works. (ii) In structural causal model view, CausalDA aims to discover causal factors instead of indirectly removing the non-causal ones. (iii) CausalDA builds the structural causal model at the logits level, whilst previous methods construct on raw data.

## 3 Methodology

### 3.1 Unified SFDA problem

SFDA aims to transfer a model pre-trained on the source domain to a different but related target domain without labeling. Formally, let $\mathcal{X}_s = \{\boldsymbol{x}_i^s\}_{i=1}^{n_s}$ and $\mathcal{Y}_s = \{y_i^s\}_{i=1}^{n_s}$ be the source samples and their truth labels. Similarly, the unlabeled samples and the truth target labels are $\mathcal{X} = \{\boldsymbol{x}_i\}_{i=1}^{n}$ and $\mathcal{Y} = \{y_i\}_{i=1}^{n}$, respectively. The unified SFDA task is to learn a target model $f_{\theta_t} : \mathcal{X} \to \mathcal{Y}$ *without any prior knowledge of the target domain not the relationships between the source and target domains (see Tab. 1)*, given (i) the source model $f_{\theta_s}$ pre-trained over $\mathcal{X}_s$, $\mathcal{Y}_s$, and (ii) the unlabeled target data $\mathcal{X}$.

### 3.2 Causality in transfer learning with source domain

In transfer learning with available source domain, such as DG and UDA, the causality relation hidden in the statistical dependence (between raw image $\boldsymbol{x}$ and available label $\boldsymbol{y}$) can be summarized to a Structural Causal Model (SCM). As depicted in Fig. 1(a), $\boldsymbol{x}$ is generated jointly by non-causal factors $U$ and causal factors $S$, while label $\boldsymbol{y}$ is determined by $S$ alone. Here, non-causal factors collectively represent the latent variables that generate the category-independent appearance of images, e.g., different domain styles and spurious dependence (Fig. 2); The causal factors present the ones determining the classification, e.g., the object shape.

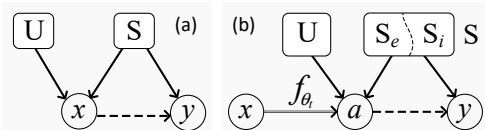

Figure 1: Differences of Structural Causal Models (SCM). **(a)** SCM for transfer learning with source domain, where $S$ and $U$ represent causal and non-causal factors. Domain shift is generally caused by $U$, which, together with the generalizable information $S$, e.g., the shape/structure of a dog (see Fig. 2), form the observation $x$. **(b)** Our proposed SCM for SFDA in a latent space (e.g., the logit $\boldsymbol{a}$) where the causal factors $S$ are decomposed into the external $S_e$ and internal $S_i$ components.

In a causal view, solving the transfer learning involves eliminating the domain shift by building a robust classification function, i.e., $P(\boldsymbol{y}|S)$. In the view of SCM, we need to disconnect $U$ to $\boldsymbol{x}$. So existing works impose intervention upon $\boldsymbol{x}$, according to the intervention theory (Pearl, 2009). Formally, this scheme can be formulated as the following form.

$$P(\boldsymbol{y}|S) = P(\boldsymbol{y}|do(x), S), \tag{1}$$

where $do(\cdot)$ means the do-operation of imposing intervention upon the input variables.

In practice, the key idea of tackling Eq. (2) is that when we change the probability distribution of non-causal factors (intervention) whilst keeping the category (prediction consistency), the causality can be extracted. Often, domain-related data augmentation is adopted as an intervention, whilst the source labels ensure prediction consistency. However, both conditions are unavailable in SFDA, making it inapplicable.

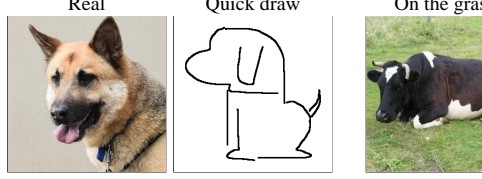

| Real | Quick draw | On the grass | On the car |

Figure 2: Illustration of domain shift. Left: **Different appearance styles** of dog; Right: **Spurious dependence of background:** Cow in grass ground vs. on a car.

### 3.3 Latent causality hypothesis for unified SFDA

To address the challenges outlined above, we propose an approach named *Causality-inspired latent invariant factor learning* (CausalDA). This method is built upon our latent causality hypothesis for unified SFDA, which explicitly conceptualizes the following core considerations. Instead of extracting causality in the raw input space $\boldsymbol{x}$, we shift our focus to the logit space $\boldsymbol{A}$: $\boldsymbol{a} = f_{\theta_t}(\boldsymbol{x}) \in \boldsymbol{A}$. The choice of this space for causality analysis is crucial for two main reasons: First and foremost, from a causal perspective, using a post-intervened space enhances causality capture by incorporating a probability distribution shift into the data (Mahajan et al., 2021; Wang et al., 2022b; Pearl, 2009). In our case, the logit space can be plausibly viewed as a post-intervened space. This interpretation stems from the inherent distributional shift between the target domain and the in-training target model (initialized via the pre-trained source model), thereby providing a reasonable justification for our use of logits.

Furthermore, the logit space is both highly semantic and compact, making it more efficient to manipulate. Since logits encode rich semantic information about class relationships, operating in this space allows for better generalization and transferability across domains. Additionally, its compact representation reduces computational complexity, facilitating more efficient learning and adaptation. These properties collectively make the logit space a more effective choice for our approach. Tab. 2 presents an analogy of CausalDA and the existing causal learning approach.

Table 2: Analogy of causality-inspired learning paradigms: existing methods, e.g., (Chen & Bühlmann, 2021; Christiansen et al., 2021; Wang et al., 2022b), versus our CausalDA.

| Approach | Intervention | Consistency |
|---|---|---|
| Existing methods | Data augmentation | Supervised |
| **CausalDA** | Using logit | Unsupervised |

An overview of CausalDA framework is illustrated in Fig. 1(b). It models both non-causal factors $U$ and causal factors $S$ within a latent space $A$. Notably, the causal component $S$ is further decomposed into internal $(S_i)$ and external $(S_e)$ sub-components to facilitate a more complete and interpretable modeling of causality.

This internal-external causality decomposition is motivated by the recognition that identifying all relevant causal relationships solely from the source and target domains is often infeasible due to limited training data and domain-specific knowledge. To address this, we introduce a pretrained ViL model, such as CLIP, as an external knowledge source to aid in the discovery of causal relationships. This design leads to two types of causality: (i) **Internal causality**, which refers to causal factors that can be readily inferred from the source domain model and target domain data; and (ii) **External causality**, which denotes the causal relations that are nearly intractable to detect from target domain observations alone, thereby requiring an external semantic prior (e.g., a ViL model with broad, many-domain information) to be uncovered. Such a collaborative design enables a more holistic understanding of causal structures, as empirically validated in `Sec.4.2` and `Sec.4.7.2`.

Unlike conventional intervention-driven methods, we focus on isolating $S$ from the entangled $U$ as

$$P(\boldsymbol{y}|S) = P(\boldsymbol{y}|dis(S|\{U, S\})), \ S = \{S_i, S_e\}, \tag{2}$$

where $dis(S|\{U, S\}))$ means conducting discover operation on $\{U, S\}$ to decompose $S$.

To make this process computationally tractable, we consider

**Principle 1** *Independent Causal Mechanisms (ICM) Principle (Peters et al., 2017): The conditional distribution of each variable given its causes (i.e., its mechanism) does not inform or influence the other mechanisms.*

Inspired by Principle 1, we assume conditional independence between external and internal causal factors to enable a divide-and-conquer learning approach. Under this perspective, we propose the following factorized approximation as:

$$P(\boldsymbol{y}|S) \approx P(\boldsymbol{y}|dis(S_e|\{U, S_e\})) \cdot P(\boldsymbol{y}|dis(S_i|\{U, S_i\})). \tag{3}$$

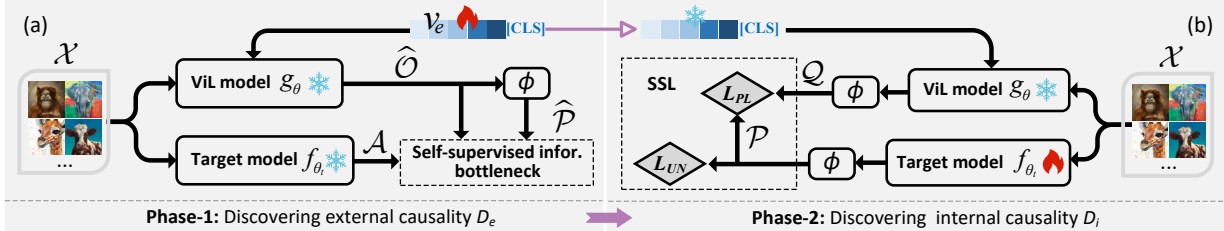

Figure 3: Overview of our CausalDA framework: (a) `Phase-1`: Discovering the external invariant factors $D_e$ in form of prompt context $\boldsymbol{v}_e$ from a frozen ViL model using our self-supervised invariant information maximization algorithm; (b) `Phase-2`: Discovering the internal invariant factors $D_i$ where the updated prompt context $\boldsymbol{v}_e$ is used to predict pseudo-labels as the prior information.

**Remark.** As analyzed in Sec. 3.2, the lack of access to real supervisions (labeled source domains and real target labels) precludes the utilization of conventional causal learning paradigms. Consequently, standard causal operations, such as establishing causal identifiability, executing true interventions, and recovering ground-truth causal factors, become intractable, making strict mathematical causality discovery for $S_e$ and $S_i$ unachievable. Instead, we shift our focus to learning latent invariant factors, which serve as an empirical proxy for causality. This choice is justified by the fact that invariant factors are inherently cross-domain-insensitive, sharing the same fundamental property as causality itself.

### 3.4 CausalDA design

Inspired by the internal-external causality hypothesis detailed above, we present a concrete design to carry out CausalDA, as depicted in Fig. 3. By Eq. (3), we implement the causality discovery in two successive phases: `phase-1` for external invariant factors $D_e$ (a proxy of $S_e$) and `phase-2` for internal invariant factors $D_i$ (a proxy of $S_i$), as detailed below.

### 3.4.1 External invariant factor discovery

To obtain external causality for a target task, we explore the potential of recent large multimodal ViL models such as CLIP (Radford et al., 2021). This is because they have accumulated a vast amount of knowledge and exhibited strong generalization abilities across a wide range of perception tasks.

Specifically, we accomplish this idea by encoding $D_e$ into the prompt context $\boldsymbol{v}_e$ of a ViL model $g_\theta$ efficiently. That is, our learning target is $\boldsymbol{v}_e$, which is the explicit expression of $D_e$. We start by formulating the $D_e$ discovery as maximizing the correlation between a random variable representing $D_e$ and the prediction of the target model. Then, the original formulation is converted to a self-supervised invariant information maximization problem with a theoretical guarantee, followed by the deep learning instantiation.

With the target samples $\mathcal{X}$ and learnable $\boldsymbol{v}_e$, we obtain the prediction of $g_\theta$ as $\hat{\mathcal{P}} = \{\hat{\boldsymbol{p}}_i\}_{i=1}^n$. Meanwhile, we obtain the logits of target model $f_{\theta_t}$ for the same data as $\mathcal{A} = \{\boldsymbol{a}_i\}_{i=1}^n$. We introduce three *random variables*, $V$, $Y$, and $Z$, following the probability distributions of $\boldsymbol{v}_e$, $\hat{\mathcal{P}}$ and $\mathcal{A}$, respectively. In CausalDA, we further write $Z = \{Z_e, Z_i\}$ where the latent random variables $Z_e$ and $Z_i$ represent the external $D_e$ and the internal $D_i$ invariant factors, respectively.

In an information-theoretic view, for causality relationships, we have to ensure the correlation between $Z_e$ and $V$. This could be considered an approximation with computational ease. Formally, discovering $D_e$ needs maximizing the following mutual information:

$$\max_{\boldsymbol{v}_e} I\left(Z_e, V\right), \tag{4}$$

where $I(\cdot, \cdot)$ is mutual information of two random variables. **Note**, mutual information is not a prerequisite for causality discovery; instead, it is an effective design choice that enhances the discovery of external causality.

To achieve Eq. (4), we employ prompt learning (Zhou et al., 2022b) to accomplish the optimization. This is because it allows us to leverage CLIP without tuning model parameters. This avoids potential issues such as catastrophic forgetting and unintended distortion of the pretrained knowledge within the ViL model. Moreover, by preserving CLIP's pretrained knowledge, prompt learning facilitates the discovery of external causality, enabling more effective integration of external knowledge into the causal learning process.

Consider that prompt $V$ is specific to the ViL model $g_\theta$, we further transform the above formula of Eq. (4) as

$$\max_{\boldsymbol{v}_e} I\left(Z_e, Y\right), Y = \phi\left(g_\theta\left(V\right)\right), \tag{5}$$

where $\phi(\cdot)$ is softmax function; so that the two variables under correlation maximization reside in similar semantic spaces. Theoretically, we prove that this transformation can constitute a lower bound on the original, ensuring its validity in optimization.

First, we have the following **Lemma** 1 (see the proof in `Appendix-A`).

**Lemma 1** *Given random variables $Z_1$, $X_1$ and $Y_1$ where $X_1$, $Y_1$ satisfy a mapping $f_1 : X_1 \mapsto Y_1$. When $f_1$ is compressed, i.e., the output's dimension is smaller than the input's,*

$$I\left(Z_1, X_1\right) \geq I\left(Z_1, Y_1\right). \tag{6}$$

The ViL model $g_\theta$ is indeed compressing, as it maps a high-dimensional image to a low-dimensional category vector. Let $Z_1 = Z_e$, $X_1 = X$, $Y_1 = Y$ and $f_1 = \phi\left(g_\theta(\cdot)\right)$, we have $I(Z_e, V) \geq I(Z_e, Y)$ according to **Lemma** 1.

**Self-supervised invariant information maximization.** In Eq. (5), $Z_e$ is the hidden part of the observable logits along with non-causal and internal invariant factors. To facilitate this learning process of $Z_e$, we derive a self-supervised causal information maximization algorithm without prior assumption on the distribution. Specifically, we introduce a random variable $Z'$ for the ViL's logits $\widehat{\mathcal{O}} = \{\hat{\boldsymbol{o}}_i\}_{i=1}^n$ between $V$ and $Y$, forming a dual information maximization expression as:

$$\max_{\boldsymbol{v}_e} I\left(Z, Z'\right) + I\left(Z', Y\right), \tag{7}$$

where $Z$ is the logit variable of target model $f_{\theta_t}$ which is observed. In Eq. (7), the first term aligns the generic external knowledge of the ViL model ($Z'$) with the target domain representations ($Z$), ensuring that the extracted $Z_e$ is specifically tailored to the target domain. The second term is in a self-supervised formulation, as $Y = \phi(Z')$ is obtained by applying a softmax operation to $Z'$. However, directly maximizing $I(Z', Y)$ is insufficient for true causal discovery. Due to the deterministic nature of this mapping $Y = \phi(Z')$, the maximization predominantly forces the model to produce low-entropy outputs. Thus, although it preserves task discriminability, it cannot distinguish whether such high-confidence predictions stem from invariant causal semantics or brittle, domain-specific artifacts, such as spurious backgrounds.

With concern on $I(Z', Y)$ mentioned above, we theoretically prove that Eq. (7) admits an lower bound, thus optimizationally valid.

**Theorem 1** *Suppose that there are five random variables $Z$, $V$, $Z'$, $Y$ and $Y'$. Among them, $Z$ represents the target domain knowledge; $V$, $Z'$ and $Y$ express the input prompt, intermediate features of the ViL model, and predictions, respectively; $Y'$ depicts a pseudo-label that has a learnable functional relationship $f_w$ ($w$ is parameters) with $Z'$, i.e., $Y' = f_w(Z')$. For the mutual information maximization objective, we have the following lower bound:*

$$\max_{\boldsymbol{v}_e} I\left(Z, Z'\right) + I\left(Z', Y\right) \geq \max_{\boldsymbol{v}_e, w} \underbrace{I\left(Z, Z'\right)}_{T_1} + \underbrace{I\left(Y', Y\right), Y' = f_w\left(Z'\right)}_{T_2}. \tag{8}$$

The proof is given in `Appendix-B`. **Theorem** 1 suggests that the desired latent factors refinement is conditionally equivalent to a self-supervised causal information maximization, when pseudo-label $Y'$ is

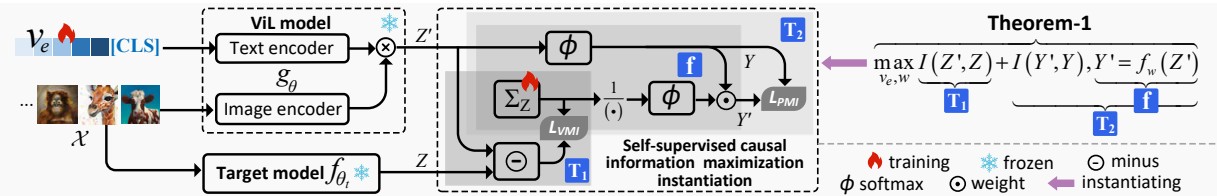

Figure 4: Realizing our self-supervised causal information maximization w.r.t. **Theorem** 1. $T_1$ is estimated by Variational Mutual Information (VMI) with Gaussian distribution assumption $\mathcal{N}(Z, \Sigma_Z)$. $T_2$ is computed by Probabilistic Mutual Information (PMI), where the function $f_w$ generates causal guidance $Y'$, employing the covariance matrix $\Sigma_Z$ from $T_1$ as weighting parameters $w$.

generated from the intermediate ViL feature $Z'$ under a mapping. We realize the proposed invariant information maximization, building on variational information maximization (Houthooft et al., 2016). Fig. 4 shows our pipeline. The instantiations of $T_1$ and $T_2$ are elaborated below.

As exactly computing mutual information of two vector variables is intractable, we maximize the variational lower bound for the term $T_1$ (see the proof in (Barber & Agakov, 2004)):

$$I\left(Z, Z'\right) = \mathbb{E}_{Z,Z'}\left[\log \frac{q(Z'|Z)}{p\left(Z'\right)}\right] + \mathrm{KL}\left(q\left(Z'|Z\right) \| p\left(Z'|Z\right)\right) \geq H\left(Z'\right) + \mathbb{E}_{Z,Z'}\left(\log q\left(Z'|Z\right)\right), \quad (9)$$

where $q(Z'|Z)$ is variational distribution approximating the real distribution $p(Z'|Z)$, $\mathrm{KL}(\cdot, \cdot)$ is KL-divergence function, and $H(Z')$ is a constant. In practice, we model $q(Z'|Z)$ as a Gaussian distribution $\mathcal{N}(Z, \Sigma_Z)$ with mean $Z$ and diagonal covariance matrix $\Sigma_Z$. For maximizing $T_1$, we adopt the Variational Mutual Information (VMI) loss (see $\mathbf{T_1}$ in Fig. 4):

$$\mathcal{L}_{\mathrm{VMI}}\left(\mathcal{X}; \boldsymbol{v}_e, \Sigma_Z\right) = \mathbb{E}_{Z,Z'}\left(\|Z' - Z\|_{\Sigma_Z^{-1}}^2 + \log |\Sigma_Z|\right) = \frac{1}{n}\sum_{i=1}^{n}(\hat{\boldsymbol{o}}_i - \boldsymbol{a}_i)^T \Sigma_Z^{-1}(\hat{\boldsymbol{o}}_i - \boldsymbol{a}_i) + \log |\Sigma_Z^i|. \quad (10)$$

Consider that the ViL's prediction $Y$ and the pseudo-label $Y'$ are probability distributions, we compute the $T_2$ term by vanilla mutual information (Ji et al., 2019): $I(Z, Z') = p(Z, Z')/[p(Z)p(Z')]$. The corresponding Probabilistic Mutual Information (PMI) loss function is formed as:

$$\mathcal{L}_{\mathrm{PMI}}\left(\mathcal{X}; \boldsymbol{v}_e, \Sigma_Z\right) = \frac{1}{n}\sum_{i=1}^{n} I\left(\hat{\boldsymbol{p}}_i, \boldsymbol{p}'_i\right), \text{with } \hat{\boldsymbol{p}}_i = \phi(\hat{\boldsymbol{o}}_i), \quad \boldsymbol{p}'_i = f_w(\hat{\boldsymbol{o}}_i), \quad (11)$$

where $\sum_{c=1}^{C} \boldsymbol{p}'_{i,c} = 1$; $f_w(\cdot)$ is a weighting-based function constructed as follows.

The key to constructing the function $f_w$ lies in satisfying three critical properties: (i) The output is a probability distribution. (ii) It is inherently an information-distilling mapping that satisfies the Data Processing Inequality (DPI), ensuring that redundant non-causal features can be eliminated. (iii) It explicitly functions as a causal filter, actively discarding unstable domain noises to isolate and distill invariant causal semantics.

In practice, the softmax projection ($\phi$) simultaneously satisfies the first two properties. It converts the logit $Z'$ into a valid probability distribution while acting as a many-to-one, non-invertible mapping. By discarding the absolute scale of the logits and compressing the feature space into a probability simplex, it enforces the information loss required by DPI, rendering $f_w$ a rigorous information-distilling mapping.

To fulfill the third property, we explicitly couple the weighting parameters $w$ to $I(Z', Z)$. Specifically, $I(Z', Z)$ functions as a Mean-Square Error weighted by $\Sigma_Z$ (see Eq. 10), inherently quantifying the domain sensitivity of the representations. High-variance dimensions reflect severe fluctuations across the target data, corresponding to non-causal domain noises ($U$) such as backgrounds. Conversely, low-variance dimensions demonstrate stability across samples, encapsulating domain-invariant causal semantics ($S_e$) like intrinsic shapes. Thus,

inverse-variance weighting tends to suppress brittle domain artifacts and amplify robust causal features. With this weighting, the resulting $Y'$ provides purified causal guidance.

Inspired by the analysis above, we take $1/\Sigma_Z$ as the weighting parameters of $f_w$, thereby shifting the focus of predicting to the causal component $S_e$ (see **f** in Fig. 4), formally,

$$\boldsymbol{p}'_i = f_{\left[ w = \Sigma^i_Z \right]} (\hat{\boldsymbol{o}}_i) = \phi \left( \phi \left( \frac{1}{\Sigma^i_Z} \right) \odot \phi \left( \hat{\boldsymbol{o}}_i \right) \right), \tag{12}$$

where $\phi(\cdot)$ is softmax function. Combining Eq. (10), (11), and (12) together, our self-supervised objective for external causality discovery is formed as

$$\max_{\boldsymbol{v}_e, \Sigma_Z} \mathcal{L}_{\text{EC}} \left( \mathcal{X}; \boldsymbol{v}_e, \Sigma_Z \right) = \mathcal{L}_{\text{PMI}} \left( \mathcal{X}; \boldsymbol{v}_e, \Sigma_Z \right) + \alpha \mathcal{L}_{\text{VMI}} \left( \mathcal{X}; \boldsymbol{v}_e, \Sigma_Z \right). \tag{13}$$

### 3.4.2 Internal invariant factor discovery

In `phase-2`, we subsequently discover the internal invariant factors $D_i$ by updating the target model. That is, we encode $D_i$ in the target model, as it is latent in nature. Similar to capturing the external factors $D_e$ (Eq. (5)), this encoding can be formulated by maximizing the following conditional mutual information:

$$\max_{\theta_t} I \left( Z_i, Z_\theta \right), \quad \text{s.t.} \quad \max_{\theta_t} I \left( Z^*_e, Z_\theta \right) \tag{14}$$

where random variable $Z_\theta$ represents target model parameters $\theta_t$, whilst $Z_i$ denotes random variable of internal invariant factors $D_i$; $Z^*_e$ represents the currently discovered external invariant factor $D^*_e$. The primary term, $\max_{\theta_t} I \left( Z_i, Z_\theta \right)$, aims to capture the internal $D_i$. The condition, $\max_{\theta_t} I \left( Z^*_e, Z_\theta \right)$, ensures the target model simultaneously accommodates the discovered external factors $D^*_e$.

Given ultra-high dimension and complexity of $\theta_t$, combined with the hidden nature of $Z_i$, directly optimizing or computing Eq. (14) is intractable. To address this, we interpret the interaction between $Z_\theta$ and $Z_i$ as the inherent modeling process that transforms an input sample from pixel space into a predictive probability space. This perspective allows us to reformulate the problem as maximizing the mutual information between the target model's input and output, both of which are *observed* and *operational*. Crucially, theoretical proofs demonstrate that mutual information between inputs and outputs is maximized when outputs are both highly confident and exhibit balanced class distributions (Krause et al., 2010). Inspired by this theoretical finding, we maximize this originally non-computable mutual information by minimizing the following objective:

$$\mathcal{L}_{\text{UN}}(\mathcal{P}; \theta_t) = - \sum_{i=1}^{n} \boldsymbol{p}_i \log \boldsymbol{p}_i + \tau \sum_{c=1}^{C} \text{KL} \left( \varrho_c \| \frac{1}{C} \right), \quad \text{with } \boldsymbol{p}_i \in \mathcal{P} = \phi(f_{\theta_t}(\mathcal{X})). \tag{15}$$

In this equation, $\mathcal{P}$ represents the predictions of $n$ training samples $\mathcal{X}$ generated by the target model $f_{\theta_t}$. The first term minimizes the information entropy of each prediction $\boldsymbol{p}_i = [p_{i,c}]_{c=1}^{C}$. The KL divergence term, on the other hand, regulates the predictive balance across all $C$ categories via pulling the empirical distribution $\varrho_c = \frac{1}{n} \sum_{i=1}^{n} p_{i,c}$ towards a uniform distribution. The trade-off parameter $\tau$ balances these two terms.

Similarly, we convert the external factors $D^*_e$ from its prompt format $\boldsymbol{v}_e$ to pseudo-labels by evaluating the ViL model $g_\theta$: $\mathcal{Q} = \phi(g_\theta(\mathcal{X}, \boldsymbol{v}_e))$. This conversion simplifies model optimization, as it enables the use of the conventional softmax cross-entropy loss, $\mathcal{L}_{\text{SCE}} (\mathcal{P}, \mathcal{Q}; \theta_t)$.

Combining these elements, we optimize Eq. (14) using the following objective function:

$$\mathcal{L}_{\text{IC}} = \min_{\theta_t} \mathcal{L}_{\text{UN}}(\mathcal{P}; \theta_t) + \sigma \mathcal{L}_{\text{SCE}} \left( \mathcal{P}, \mathcal{Q}; \theta_t \right) \tag{16}$$

where $\sigma$ serves as another trade-off parameter.

### 3.5 Model training

We train CausalDA in an alternating manner. In each iteration, we first optimize prompt context $\boldsymbol{v}_e$ with $\mathcal{L}_{\text{EC}}$ whilst the target model is frozen (i.e., `phase-1`), followed by training the target model by $\mathcal{L}_{\text{IC}}$ (i.e., `phase-2`) while the prompt context is frozen. The ViL model is always frozen throughout. More details for training are presented in Alg. 1.

---

**Algorithm 1** Training CausalDA

---

**Input**: Unlabelled target data $\mathcal{X}$, pre-trained source model $f_{\theta_s}$, a frozen ViL model $g_\theta$, max training iteration number $M$, prompt context $\boldsymbol{v}_e$. Initialize target model $f_{\theta_t}$(by $f_{\theta_s}$), $\boldsymbol{v}_e$(by fixed template)

    **for** iter $k = 1$ to $M$ **do**
        Sample a mini-batch $\mathcal{X}_b$ from $\mathcal{X}$.
        Discover external invariant factors $D_e$: update $(\boldsymbol{v}_e, \Sigma_Z)$ by optimizing $\mathcal{L}_{\text{EC}}(\mathcal{X}_b; \boldsymbol{v}_e, \Sigma_Z)$, fixing $\mathcal{L}_{\text{IC}}$.
        Convert updated $\boldsymbol{v}_e$ to target pseudo-labels for this batch $\mathcal{X}_b$, i.e., obtain $\mathcal{Q}_b = \phi(g_\theta(\mathcal{X}_b, \boldsymbol{v}_e))$.
        Discover internal invariant factors $D_i$: update $f_{\theta_t}$ by optimizing $\mathcal{L}_{\text{IC}}$, fixing $\mathcal{L}_{\text{EC}}$.
    **end for**
    **return:** The adapted model $f_{\theta_t}$.

---

## 4 Experiments

### 4.1 Implementation details

**Source models.** Following (Zhou et al., 2022b), for source-free out-of-distribution generalization (SF-OODG) we adopt the Pytorch built-in ResNet50 model (pre-trained on ImageNet) as the source model, taking ImageNet variations as evaluation datasets. For other settings, the source model is trained on the source domain in a supervised manner, same as (Liang et al., 2020; Tang et al., 2022).

**Networks.** Our CausalDA involves two parts: the ViL model and the target model. We choose CLIP (Radford et al., 2021) as the ViL model. The target model's structure is the same as the source model. Following (Liang et al., 2020; Yang et al., 2023), a target model consists of a feature extractor and a classifier (an FC layer). We adapt the feature extractor, which is pre-trained on ImageNet. For fair comparison, we use ResNet101 as the backbone for VisDA and ResNet50 for the others.

**Training.** There are three parameters with CausalDA: $\alpha$ in $L_{\text{EC}}$ (Eq. (13)), $\tau$ in $L_{\text{UN}}$ (Eq. (15)), and $\sigma$ in $L_{\text{IC}}$ (Eq. (16)). For all settings, we adopt the same configuration $(\alpha, \sigma, \tau) = (0.003, 0.4, 1.0)$. Here, since the value of $\mathcal{L}_{\text{VMI}}$ is large, $\alpha$ is set as small as 0.003. For model training, we use a batch size of 64, an SGD optimizer with momentum 0.9, and 15 epochs on all datasets. The learnable prompt template is initialized with 'a photo of a [CLS].' (Radford et al., 2021), where the [CLS] is a specific target class name.

In the implementation, [CLS] is instantiated strictly with known source class names, meaning our approach does not require access to unknown target class names in non-closed-set settings. To identify unknown target samples, we follow the uncertainty-based rejection strategy from SHOT (Liang et al., 2020). We compute the entropy of the model's prediction distribution over the known source classes and utilize KMeans clustering to separate confident, low-entropy samples from uncertain, high-entropy ones. The high-entropy cluster is then rejected as belonging to unknown categories. Thus, the detection of unknown classes is driven entirely by prediction uncertainty rather than reliance on target-private class names.

**Unification evaluation protocol**. In the SFDA context, unification does not necessarily imply that the proposed method must achieve the absolute best performance in every individual adaptation scenario. Instead, the essence of contributing a unified solution lies in its ability to provide stable and competitive performance across a wide range of settings in a unified manner, without the need of resorting to a diversity of scenario-specific designs.

To evaluate the unification of methods, we adopt three harmonic metrics: (1) Overall mean $H_{all}$ that averages over all settings, (2) worst-case relative gap $H_{wrg}$ that is largest percentage shortfall of a method from the per-setting best across all settings, and (3) leave-one-setting-out averages $H_{loso}$ omits one setting at a time and averages the scores over the rest. For a specific method $m$ evaluated over a number of settings $\mathcal{S}$, the metrics can be computed by:

$$\text{H}_{all}(m) = \frac{1}{|\mathcal{S}|} \sum_{s \in \mathcal{S}} x_{m,s}; \ \text{H}_{wrg}(m) = \max_{s \in \mathcal{S}} \frac{b_s - x_{m,s}}{b_s}; \ \text{H}_{loso}(m, s') = \frac{1}{|\mathcal{S} - 1|} \sum_{s \in \mathcal{S} \setminus s'} x_{m,s}, \quad (17)$$

where $x_{m,s}$ specifies the score of method $m$ under a setting $s$; $b_s = \max_{m'} x_{m',s}$ is the per-setting best; $s'$ is a omitted setting.

## 4.2 Indirect validation of causality based on invariance analysis

Validating the discovery of causal factors is significant yet non-trivial in causality learning, especially with latent factors, as in this study. To that end, we adopt a general principle that there exists an intimate connection between invariance and causality useful for generalization (Schölkopf et al., 2021; Arjovsky et al., 2019). That being said, if a learned model enables varying environments generalization, it constitutes *indirect evidence* that the model has captured causal predictors (e.g., a causal explanation of an object, such as why it is a computer).

Concretely, we simulate a spectrum of viewing environments by imposing varying levels of Gaussian noise with gradually increased kernel sizes from 8 to 20. We compare the competitors without (SHOT (Liang et al., 2020)) and with (DIFO (Tang et al., 2024c) and ProDe (Tang et al., 2025)) ViL knowledge (the image-encoder in CLIP adopts backbone ViT-B/32 (Han et al., 2022)). As shown in Fig. 5, all competitors degrade clearly as the noise increases, whilst CausalDA levels off. These findings suggest the method captures invariant predictors that are compatible with causal explanations, though

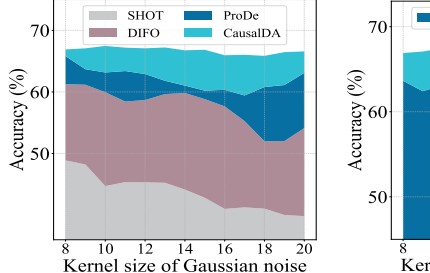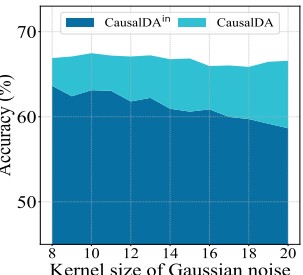

Figure 5: Causality validation: Invariance analysis across varying noise settings on the Ar→Cl task of **Office-home**.

they do not prove causal recovery rigorously. To further examine if external and internal factors both are effective, we further derive a variant of CausalDA, named as CausalDA$^{in}$, without the external invariant factor discovery. Specifically, in CausalDA$^{in}$, the pseudo-labels are directly derived from CLIP's zero-shot predictions. This is consistent with external factors contributing to improved invariance, suggesting a possible complementarity with internal counterparts.

The preceding validation provides direct yet preliminary evidence for invariant factor identification, thereby serving as indirect evidence for causal discovery. In the following sections, we present further corroborating evidence through adaptation results across diverse SFDA settings.

## 4.3 Closed-set SFDA

**Datasets**. The evaluation is based on three challenging benchmarks. **Office-Home** (Venkateswara et al., 2017) is a middle-scale dataset. It contains 15k images belonging to 65 categories from working or family environments, being divided into four domains, i.e., **Ar**tistic, **Cl**ip Art, **Pr**oduct, and **R**eal-**w**ord. **VisDA** (Peng et al., 2017) is a large-scale dataset. It includes 12 types of **S**ynthetic to **R**eal transfer recognition tasks. The source domain contains 152k synthetic images, whilst the target domain has 55k real object images from COCO. **DomainNet-126** (Peng et al., 2019) is another large-scale dataset. As a subset of DomainNet containing 600k images of 345 classes from 6 domains of different image styles, this dataset has 145k images from 126 classes, sampled from 4 domains, **C**lipart, **P**ainting, **R**eal, **S**ketch, as (Peng et al., 2019) identifies severe noisy labels in the dataset.

**Competitors**. To evaluate CausalDA, we select 14 state-of-the-art methods in three groups. *(i) The first* is the source model and CLIP that lay the basis for CausalDA. *(ii) The second* is the UDA method DAPL (Ge et al., 2023), PADCLIP (Lai et al., 2023), ADCLIP (Singha et al., 2023) that introduces prompt learning to boost the cross-domain transfer. *(iii) The third* includes 9 SFDA methods: SHOT (Liang et al., 2020), GKD (Tang et al., 2021), NRC (Yang et al., 2021a), AaD (Yang et al., 2022), AdaCon (Chen et al., 2022a), CoWA (Lee et al., 2022), PLUE (Litrico et al., 2023), TPDS (Tang et al., 2024a), and MMGA (Chen et al., 2026). By idea, SHOT, GKD, CoWA are pseudo-labels-based methods; NRC, AaD, PLUE, AdaCon exploit the data geometry, e.g., the nearest neighbor, neighborhood, prototypes; TPDS predicts the target probability

Table 3: Closed-set results (%) on **Office-Home**, **VisDA**. SF and ViL means source-free and ViL model-required, respectively.

| Method | Venue | SF | ViL | Ar→Cl | Ar→Pr | Ar→Rw | Cl→Ar | Cl→Pr | Cl→Rw | Pr→Ar | Pr→Cl | Pr→Rw | Rw→Ar | Rw→Cl | Rw→Pr | Avg. | VisDA S→R |
|---|---|---|---|---|---|---|---|---|---|---|---|---|---|---|---|---|---|
| Source | – | – | – | 43.7 | 67.0 | 73.9 | 49.9 | 60.1 | 62.5 | 51.7 | 40.9 | 72.6 | 64.2 | 46.3 | 78.1 | 59.2 | 49.2 |
| CLIP-RN | ICML21 | – | ✓ | 51.9 | 81.5 | 82.5 | 72.5 | 81.5 | 82.5 | 72.5 | 51.9 | 82.5 | 72.5 | 51.9 | 81.5 | 72.1 | 83.7 |
| CLIP-B32 | ICML21 | – | ✓ | 64.9 | 86.6 | 87.2 | 78.0 | 86.6 | 87.2 | 78.0 | 64.9 | 87.2 | 78.0 | 64.9 | 86.6 | 79.2 | 85.2 |
| DAPL-RN | TNNLS23 | ✗ | ✓ | 54.1 | 84.3 | 84.8 | 74.4 | 83.7 | 85.0 | 74.5 | 54.6 | 84.8 | 75.2 | 54.7 | 83.8 | 74.5 | 86.9 |
| PADCLIP-RN | ICCV23 | ✗ | ✓ | 57.5 | 84.0 | 83.8 | 77.8 | 85.5 | 84.7 | 76.3 | 59.2 | 85.4 | 78.1 | 60.2 | 86.7 | 76.6 | 88.5 |
| ADCLIP-RN | ICCV23W | ✗ | ✓ | 55.4 | 85.2 | 85.6 | 76.1 | 85.8 | 86.2 | 76.7 | 56.1 | 85.4 | 76.8 | 56.1 | 85.5 | 75.9 | 87.7 |
| SHOT | ICML20 | ✓ | ✗ | 55.0 | 78.7 | 81.3 | 69.1 | 78.9 | 79.1 | 68.2 | 53.6 | 81.6 | 73.5 | 59.4 | 83.5 | 71.8 | 82.9 |
| GKD | IROS21 | ✓ | ✗ | 56.5 | 78.3 | 82.2 | 69.2 | 80.4 | 78.7 | 67.4 | 55.4 | 82.6 | 74.3 | 60.3 | 84.2 | 72.5 | 83.0 |
| NRC | NeurIPS21 | ✓ | ✗ | 57.2 | 79.3 | 81.3 | 68.9 | 80.6 | 80.2 | 66.6 | 57.3 | 82.0 | 71.0 | 57.9 | 84.9 | 72.3 | 85.9 |
| AaD | NeurIPS22 | ✓ | ✗ | 59.3 | 79.3 | 82.1 | 68.9 | 79.8 | 79.5 | 67.2 | 57.4 | 83.1 | 72.1 | 58.5 | 85.4 | 72.7 | 88.0 |
| AdaCon | CVPR22 | ✓ | ✗ | 47.2 | 75.1 | 75.5 | 60.7 | 73.3 | 73.2 | 60.2 | 45.2 | 76.6 | 65.6 | 48.3 | 79.1 | 65.0 | 86.8 |
| CoWA | ICML22 | ✓ | ✗ | 57.3 | 79.3 | 81.0 | 69.3 | 77.9 | 79.6 | 68.1 | 56.4 | 82.6 | 72.9 | 61.3 | 83.7 | 72.4 | 86.9 |
| PLUE | CVPR23 | ✓ | ✗ | 49.1 | 73.5 | 78.2 | 62.9 | 73.5 | 74.5 | 62.2 | 48.3 | 78.6 | 68.6 | 51.8 | 81.5 | 66.9 | 88.3 |
| TPDS | IJCV24 | ✓ | ✗ | 59.3 | 80.3 | 82.1 | 70.6 | 79.4 | 80.9 | 69.8 | 56.8 | 82.1 | 74.5 | 61.2 | 85.3 | 73.5 | 87.6 |
| SHOT-B32 | ICML20 | ✓ | ✓ | 64.7 | 84.0 | 86.4 | 79.4 | 87.3 | 85.4 | 78.1 | 65.8 | 87.6 | 81.6 | 65.0 | 88.5 | 79.5 | 85.2 |
| GKD-B32 | IROS21 | ✓ | ✓ | 67.2 | 83.2 | 84.3 | 77.0 | 84.8 | 83.6 | 75.5 | 71.1 | 86.0 | 81.1 | 72.4 | 88.5 | 79.6 | 85.3 |
| NRC-B32 | NeurIPS21 | ✓ | ✓ | 55.0 | 83.4 | 86.7 | 78.3 | 86.5 | 85.1 | 77.1 | 69.9 | 87.2 | 90.9 | 65.5 | 88.4 | 79.5 | 88.2 |
| AaD-B32 | NeurIPS22 | ✓ | ✓ | 69.4 | 83.0 | 84.9 | 76.4 | 85.2 | 84.6 | 73.3 | 71.1 | 86.3 | 79.0 | 72.1 | 88.7 | 79.5 | 82.6 |
| TPDS-B32 | IJCV24 | ✓ | ✓ | 67.7 | 82.7 | 86.1 | 76.5 | 88.0 | 86.3 | 73.6 | 69.4 | 87.1 | 79.3 | 70.5 | 89.7 | 79.7 | 78.0 |
| MMGA-B32 | PR26 | ✓ | ✓ | 68.5 | 84.6 | 86.8 | 78.8 | 87.5 | 86.5 | 75.9 | 68.3 | 86.3 | 80.4 | 69.9 | 88.6 | 80.2 | 89.2 |
| **CausalDA**-C-RN | – | ✓ | ✓ | 60.1 | 85.6 | 86.2 | 77.2 | 86.0 | 86.3 | 76.6 | 61.0 | 86.5 | 77.5 | 61.4 | 86.2 | 77.6 | 89.3 |
| **CausalDA**-C-B32 | – | ✓ | ✓ | **74.2** | **90.8** | **90.3** | **82.3** | **91.0** | **89.9** | **81.0** | **74.8** | **90.0** | 82.5 | **74.1** | **90.9** | **84.3** | **90.3** |

distribution using error control; MMGA aligns the target domain with the multimodal space represented by CLIP.

For comprehensive comparisons, we implement two variants: (i) CausalDA-C-RN (base) and (ii) CausalDA-C-B32 (premium). The key distinction lies in the backbone of the CLIP image encoder. Specifically, for CausalDA-C-RN, ResNet101 is employed on the VisDA dataset, while ResNet50 is used on the Office-Home and DomainNet-126 datasets. On the other hand, CausalDA-C-B32 adopts ViT-B/32 (Han et al., 2022) as the backbone across all datasets. This extends to all other methods using CLIP.

For an SFDA-focused comparison, we integrate the same CLIP model with existing SFDA methods, utilizing the CLIP's visual backbone (ViT B/32) as the feature extractor, followed by the original training algorithm. As such, all compared methods benefit from the CLIP pre-trained knowledge in a design-generic manner. Specifically, we select five updated SFDA methods as comparisons, including SHOT-B32, GKD-B32, NRC-B32, AaD-B32, and TPDS-B32.

**Results.** As listed in Tab. 3∼4, our CausalDA-C-RN and CausalDA-C-B32 perform best on all tasks across the three datasets except for two tasks. Compared with previous best SFDA methods TPDS (on Office-Home), PLUE (on VisDA) and GKD (on DomainNet-126), CausalDA-C-RN improves by **4.1**%, **1.0**% and **9.3**% in average accuracy, respectively. Compared with these UDA methods with both access to labelled source data and CLIP, CausalDA-C-RN still improves by **1.0**%, **0.8**%, and **2.8**% on top of second-best methods PADCLIP-RN (on Office-Home, VisDA) and ADCLIP-RN (on DomainNet-126). The improvement expands further when we switch focus to the strong version CausalDA-C-B32. Such results are unsurprising since the well-pretrained CLIP is employed to identify the external invariant elements.

On the other hand, CausalDA-C-RN defeats CLIP-RN on all tasks, achieving the improvement of **4.5**%, **5.6**%, and **5.3**% on Office-Home, VisDA, and DomainNet-126 on average, respectively. Similarly, compared with CLIP-B32, the corresponding improvement of CausalDA-C-B32 are changed to **5.1**%, **5.1**% and **3.0**%. This indicates the effect of the prompt learning to discover the external elements.

Among the "-B32" models, we have two observations. First, CausalDA-C-B32 outperforms all conventional methods across all tasks on the three datasets, confirming the effectiveness of our design. Second, on the VisDA dataset, improvements with the B32 versions are limited, and they are even outperformed by the ResNet version, such as AaD and TPDS. This is because the rendered synthetic data (the source domain in VisDA) lacks visual details, appearing primarily in shades of white and gray light and shadow. When

Table 4: Closed-set results (%) on **DomainNet-126**. SF, ViL means: source-free, ViL model-required.

| Method | Venue | SF | ViL | C→P | C→R | C→S | P→C | P→R | P→S | R→C | R→P | R→S | S→C | S→P | S→R | Avg. |
|---|---|---|---|---|---|---|---|---|---|---|---|---|---|---|---|---|
| Source | – | – | – | 44.6 | 59.8 | 47.5 | 53.3 | 75.3 | 46.2 | 55.3 | 62.7 | 46.4 | 55.1 | 50.7 | 59.5 | 54.7 |
| CLIP-RN | ICML21 | – | ✓ | 70.2 | 87.1 | 65.4 | 67.9 | 87.1 | 65.4 | 67.9 | 70.2 | 65.4 | 67.9 | 70.2 | 87.1 | 72.7 |
| CLIP-B32 | ICML21 | – | ✓ | 76.2 | 89.0 | 73.6 | 77.9 | 89.0 | 73.6 | 77.9 | 76.2 | 73.6 | 77.9 | 76.2 | 89.0 | 79.2 |
| DAPL-RN | TNNLS23 | ✗ | ✓ | 72.4 | 87.6 | 65.9 | 72.7 | 87.6 | 65.6 | 73.2 | 72.4 | 66.2 | 73.8 | 72.9 | 87.8 | 74.8 |
| ADCLIP-RN | ICCV23W | ✗ | ✓ | 71.7 | 88.1 | 66.0 | 73.2 | 86.9 | 65.2 | 73.6 | 73.0 | 68.4 | 72.3 | 74.2 | 89.3 | 75.2 |
| SHOT | ICML20 | ✓ | ✗ | 63.5 | 78.2 | 59.5 | 67.9 | 81.3 | 61.7 | 67.7 | 67.6 | 57.8 | 70.2 | 64.0 | 78.0 | 68.1 |
| GKD | IROS21 | ✓ | ✗ | 61.4 | 77.4 | 60.3 | 69.6 | 81.4 | 63.2 | 68.3 | 68.4 | 59.5 | 71.5 | 65.2 | 77.6 | 68.7 |
| NRC | NeurIPS21 | ✓ | ✗ | 62.6 | 77.1 | 58.3 | 62.9 | 81.3 | 60.7 | 64.7 | 69.4 | 58.7 | 69.4 | 65.8 | 78.7 | 67.5 |
| AdaCon | CVPR22 | ✓ | ✗ | 60.8 | 74.8 | 55.9 | 62.2 | 78.3 | 58.2 | 63.1 | 68.1 | 55.6 | 67.1 | 66.0 | 75.4 | 65.4 |
| CoWA | ICML22 | ✓ | ✗ | 64.6 | 80.6 | 60.6 | 66.2 | 79.8 | 60.8 | 69.0 | 67.2 | 60.0 | 69.0 | 65.8 | 79.9 | 68.6 |
| PLUE | CVPR23 | ✓ | ✗ | 59.8 | 74.0 | 56.0 | 61.6 | 78.5 | 57.9 | 61.6 | 65.9 | 53.8 | 67.5 | 64.3 | 76.0 | 64.7 |
| TPDS | IJCV24 | ✓ | ✗ | 62.9 | 77.1 | 59.8 | 65.6 | 79.0 | 61.5 | 66.4 | 67.0 | 58.2 | 68.6 | 64.3 | 75.3 | 67.1 |
| SHOT-B32 | ICML20 | ✓ | ✓ | 72.8 | 84.5 | 72.0 | 75.1 | 85.2 | 72.2 | 78.2 | 75.5 | 71.6 | 75.2 | 73.6 | 83.2 | 76.6 |
| GKD-B32 | IROS21 | ✓ | ✓ | 73.7 | 84.4 | 73.6 | 76.1 | 85.4 | 73.6 | 80.4 | 76.9 | 73.9 | 74.9 | 74.8 | 82.1 | 77.5 |
| NRC-B32 | NeurIPS21 | ✓ | ✓ | 72.2 | 83.4 | 72.5 | 75.1 | 84.8 | 72.6 | 78.3 | 76.7 | 74.0 | 75.0 | 73.7 | 82.6 | 76.7 |
| AaD-B32 | NeurIPS22 | ✓ | ✓ | 72.2 | 84.2 | 72.0 | 75.3 | 85.3 | 72.6 | 78.2 | 77.4 | 73.6 | 74.8 | 74.7 | 82.8 | 76.9 |
| TPDS-B32 | IJCV24 | ✓ | ✓ | 72.5 | 82.8 | 71.8 | 77.7 | 84.7 | 72.9 | **81.9** | 76.5 | 72.8 | 73.5 | 74.2 | 83.0 | 77.0 |
| MMGA-B32 | PR26 | ✓ | ✓ | 75.4 | 86.5 | 69.9 | 74.6 | 88.3 | 72.3 | 75.9 | 76.1 | 70.9 | 76.2 | 75.6 | 87.4 | 77.4 |
| **CausalDA**-C-RN | – | ✓ | ✓ | 75.4 | 88.2 | 72.0 | 75.8 | 88.3 | 72.1 | 76.1 | 75.6 | 71.2 | 77.6 | 75.9 | 88.2 | 78.0 |
| **CausalDA**-C-B32 | – | ✓ | ✓ | **79.5** | **89.8** | **76.9** | **81.6** | **90.1** | **77.5** | 81.7 | **80.4** | **76.6** | **82.5** | **80.1** | **89.9** | **82.2** |

processed in ViT, this results in confused semantics. As for AaD and TPDS, contrastive computation (AaD) and chain-like search (TPDS) amplify the noise in the confused semantics, leading to uncontrolled error propagation.

## 4.4 Generalized SFDA

**Dataset**. We evaluate CausalDA on Office-Home, following previous Generalized SFDA works (Yang et al., 2021b; Tang et al., 2024b).

**Evaluation protocol.** Unlike Closed-set SFDA, the Generalized SFDA problem highlights the anti-forgetting ability on the seen source domain. In terms of evaluation rule, the same as (Yang et al., 2021b), we adopt the harmonic mean accuracy that is computed by $H = (2 * A_s * A_t)/(A_s + A_t)$ where $A_s$ and $A_t$ are the accuracy on the source domain and the target domain, respectively. Note that the $A_s$ is computed based on the source-testing set. The same as (Yang et al., 2021b; Tang et al., 2024b), on the source domain, the ratio of training and testing sets is 9:1.

**Competitors.** As listed in Tab. 5, two Generalized SFDA comparisons, GDA (Yang et al., 2021b) and PSAT (Tang et al., 2024b), are additionally selected. GDA integrated domain attention with a local clustering-based self-supervised learning method. Essentially, knowledge of old tasks is preserved by constraining parameters near their original values. Instead, PSAT enforces the adaptation process to remember the source domain by imposing source

Table 5: One step adapting results (%) on **Office-Home** in Generalized SFDA setting. S, T, H are the results on the source and target domains, and harmonic mean accuracy, respectively; the bracket values in red are the gap between S and T; WAD means With Anti-forgetting Design.

| Method | Venue | WAD | Avg. S | T | H |
|---|---|---|---|---|---|
| Source | – | ✗ | 98.1 | 59.2 | 73.1 |
| SHOT | ICML20 | ✗ | 84.2 | 71.8 | 77.1 |
| GKD | IROS21 | ✗ | 86.8 | 72.5 | 78.7 |
| NRC | NeurIPS21 | ✗ | 91.3 | 72.3 | 80.4 |
| AdaCon | CVPR22 | ✗ | 88.2 | 65.0 | 74.4 |
| CoWA | ICML22 | ✗ | 91.8 | 72.4 | 80.6 |
| PLUE | CVPR23 | ✗ | 96.3 | 66.9 | 78.4 |
| TPDS | IJCV24 | ✗ | 83.8 | 73.5 | 78.0 |
| GDA | ICCV21 | ✓ | 80.0 | 70.2 | 74.4 |
| PSAT-RN | TMM24 | ✓ | 85.3 | 72.6 | 78.4 |
| SHOT-B32 | ICML20 | ✗ | 81.9 | 79.5 | 80.7 |
| GKD-B32 | IROS21 | ✗ | 89.2 | 79.5 | 84.1 |
| NRC-B32 | NeurIPS21 | ✗ | 81.1 | 78.6 | 79.8 |
| TPDS-B32 | IJCV24 | ✗ | 88.9 | 79.7 | 84.1 |
| PSAT-B32 | TMM24 | ✓ | 89.5 | 80.7 | 84.9 |
| MMGA-B32 | PR26 | ✗ | 86.9 | 80.2 | 83.4 |
| **CausalDA**-C-RN | – | ✗ | 85.0 | 77.6 | 80.7 |
| **CausalDA**-C-B32 | – | ✗ | 86.1 | 84.3 | **85.2** |

guidance, building a target domain-centric anti-forgetting mechanism. As CausalDA, we make two PSAT variants with ResNet50 (PSAT-RN) and ViT (PSAT-B32).

Table 6: Continual adaptation results (%) on **Office-Home** under Generalized SFDA setting, the first column of each sub-table indicates the adaptation sequence. ↓ means the average accuracy drop of a test domain on the adaptation path compared with the performance when the domain is first seen.

CausalDA-C-RN

| | Test | | | | | Test | | | | | Test | | | | | Test | | | |
|---|---|---|---|---|---|---|---|---|---|---|---|---|---|---|---|---|---|---|---|
| | Ar | Cl | Pr | Rw | | Cl | Ar | Pr | Rw | | Pr | Ar | Cl | Rw | | Rw | Ar | Cl | Pr |
| Ar | 97.8 | 46.1 | 68.4 | 71.1 | Cl | 97.5 | 52.5 | 65.8 | 61.5 | Pr | 99.6 | 52.2 | 40.8 | 71.3 | Rw | 98.1 | 63.1 | 46.3 | 78.5 |
| Cl | 80.0 | 63.5 | 64.3 | 66.0 | Ar | 80.1 | 77.8 | 70.7 | 75.6 | Ar | 91.2 | 78.4 | 50.2 | 78.3 | Ar | 93.4 | 78.4 | 50.4 | 75.6 |
| Pr | 78.4 | 61.7 | 85.0 | 76.2 | Pr | 75.4 | 74.4 | 86.9 | 78.1 | Cl | 81.5 | 71.9 | 61.5 | 69.9 | Cl | 83.0 | 74.7 | 62.1 | 69.7 |
| Rw | 83.1 | 60.2 | 83.2 | 84.4 | Rw | 67.6 | 78.4 | 85.4 | 85.6 | Rw | 87.3 | 76.9 | 59.2 | 85.6 | Pr | 87.3 | 72.5 | 61.5 | 85.2 |
| ↓ | 17.3 | 2.5 | 1.8 | − | ↓ | 23.1 | 1.4 | 1.6 | − | ↓ | 13.0 | 4.1 | 2.3 | − | ↓ | 10.2 | 4.8 | 0.6 | − |

CausalDA-C-B32

| | Test | | | | | Test | | | | | Test | | | | | Test | | | |
|---|---|---|---|---|---|---|---|---|---|---|---|---|---|---|---|---|---|---|---|
| | Ar | Cl | Pr | Rw | | Cl | Ar | Pr | Rw | | Pr | Ar | Cl | Rw | | Rw | Ar | Cl | Pr |
| Ar | 97.8 | 46.1 | 68.4 | 71.1 | Cl | 97.5 | 52.5 | 65.8 | 61.5 | Pr | 99.6 | 52.2 | 40.8 | 71.3 | Rw | 98.1 | 63.1 | 46.3 | 78.5 |
| Cl | 84.1 | 77.0 | 69.3 | 72.5 | Ar | 80.9 | 83.4 | 72.3 | 77.0 | Ar | 90.0 | 82.5 | 49.4 | 80.1 | Ar | 94.0 | 82.8 | 51.2 | 76.8 |
| Pr | 82.5 | 70.9 | 91.0 | 81.1 | Pr | 74.2 | 78.1 | 90.4 | 82.2 | Cl | 83.6 | 75.6 | 76.2 | 76.2 | Cl | 85.2 | 73.8 | 76.2 | 72.5 |
| Rw | 86.3 | 67.6 | 88.3 | 87.7 | Rw | 69.5 | 81.3 | 88.9 | 89.1 | Rw | 89.8 | 79.1 | 69.5 | 87.7 | Pr | 89.5 | 76.9 | 71.5 | 90.0 |
| ↓ | 13.5 | 7.7 | 2.7 | − | ↓ | 22.6 | 3.7 | 1.5 | − | ↓ | 11.8 | 5.2 | 6.7 | − | ↓ | 8.5 | 7.5 | 4.7 | − |

**One step adapting**. Tab. 5 reports the results, including source and target domain accuracies and their average. Among the "-B32" methods, CausalDA-C-B32 (without any anti-forgetting design) defeats the best specially designed model PSAT-B32 (with an anti-forgetting mechanism) with a lead of **0.3**% in harmonic mean accuracy. CausalDA-C-RN outperforms other alternatives with the same backbone. These results verify the competitiveness of our models in one-step adaptation.

**Continual adapting results.** We conduct continual adaptation testing following (Yang et al., 2021b). In the adaptation sequence Ar→Cl→Pr→Rw, we first train an initial model on domain Ar, which then serves as the source for adaptation to domain Cl. This process repeats iteratively until the final adaptation to domain Rw is achieved, resulting in four distinct models. Each model is evaluated across all domains, with **10**% of the data in each domain randomly selected as the test set.

Tab. 6 presents the results of CausalDA-C-RN and CausalDA-C-B32. We highlight two observations with CausalDA-C-B32: **First**, if a domain has been encountered during continual adaptation, the adapted model does not degrade significantly on that domain. For example, in the adaptation sequence Cl→Ar→Pr→Rw, when adapting from Cl to Ar, the accuracy on Ar reaches **83.4**%. Subsequent adaptations to Pr and Rw lead to an average performance drop of **3.7**% (see the second column in the second sub-table). **Second**, the adapted model performs well even on unseen domains. For instance, along the adaptation path Ar→Cl→Pr→Rw, the model's accuracy on Rw gradually improves from **71.1**% to **87.7**% (see the fifth column in the first sub-table). These results suggest that CausalDA effectively captures invariant factors, making it robust to domain shifts.

Table 7: Partial-set, Open-set results (%) on **Office-Home**.

| Partial-set SFDA | | | Open-set SFDA | | |
|---|---|---|---|---|---|
| Method | Venue | Avg. | Method | Venue | Avg. |
| Source | − | 62.8 | Source | − | 46.6 |
| SHOT | ICML20 | 79.3 | SHOT | ICML20 | 72.8 |
| HCL | NeurIPS21 | 79.6 | HCL | NeurIPS21 | 72.6 |
| CoWA | ICML22 | 83.2 | CoWA | ICML22 | 73.2 |
| AaD | NeurIPS22 | 79.7 | AaD | NeurIPS22 | 71.8 |
| CRS | CVPR23 | 80.6 | CRS | CVPR23 | 73.2 |
| SHOT-B32 | ICML20 | 80.6 | SHOT-B32 | ICML20 | 73.3 |
| AaD-B32 | NeurIPS22 | 81.2 | AaD-B32 | NeurIPS22 | 72.4 |
| **CausalDA**-C-RN | − | 82.9 | **CausalDA**-C-RN | − | 79.6 |
| **CausalDA**-C-B32 | − | **87.1** | **CausalDA**-C-B32 | − | **84.0** |

## 4.5 Open-set & Partial-set SFDA

**Dataset**. We adopt Office-Home for evaluation (Liang et al., 2020; Zhang et al., 2023).

**Competitors.** Except Source, CLIP, SHOT, and CoWA as in the Closed-set setting, we consider two more contrastive learning based SFDA methods, HCL (Huang et al., 2021) and CRS (Zhang et al., 2023).

**Comparison results.** As reported in Tab. 7, CausalDA-C-B32 achieves the highest average accuracy, outperforming all competing methods by margins of **5.9**% and **10.7**% over AaD-B32 in the Partial-set setting and SHOT-B32 in the Open-set setting, respectively. Similarly, it significantly outperforms SHOT-B32 and AaD-B32 under both Partial-set and Open-set settings, indicating the efficacy of the proposed design. Only CoWA beats CausalDA-C-RN in the Partial-set setting with a tiny lead **0.3**%. The results indicate that CausalDA is competitive for the Partial-set and Open-set settings.

## 4.6 Source-free out-of-distribution generalization

**Datasets**. We evaluate CausalDA in the SF-OODG setting on four challenging **ImageNet Variants**. Specifically, **IN-V2** (i.e., ImageNet-V2) (Recht et al., 2019) is an independent set consisting of natural images, collected from different sources, a total of 10k images of 1000 ImageNet categories. **IN-A** (i.e., ImageNet-A) (Hendrycks et al., 2021b) is a challenging set consisting of these "natural adversarial examples" (misclassified by a vanilla ResNet50 (He et al., 2016)), containing 7.5k images of 200 ImageNet categories. **IN-R** (i.e., ImageNet-R) (Hendrycks et al., 2021a) is an ImageNet sub-set with artistic renditions. It contains 30k images covering 200 ImageNet categories. **IN-K** (i.e., ImageNet-Sketch) (Wang et al., 2019) collects black and white sketches from the ImageNet validation set. This dataset contains 50k images from 1000 ImageNet categories.

**Competitors.** In addition to the methods compared in the Closed-set SFDA setting, we include four more: CoOP (Zhou et al., 2022b), CoCoOp (Zhou et al., 2022a), TPT (Shu et al., 2022), and ProGrad (Zhu et al., 2023). Among them, CoOP, CoCoOp, and ProGrad employ prompt learning with supervision from a few labeled samples, while TPT introduces self-supervised prompt tuning by minimizing multi-view entropy.

**Comparison results**. Tab. 8 reports the SF-OODG results. In average accuracy, our methods, CausalDA-C-NR and CausalDA-C-B32, outperform previous SFDA methods by a significant margin, achieving at least a **10**% improvement. Even compared to CLIP and other CLIP-based methods, CausalDA maintains a clear advantage. For example, CausalDA-C-NR improves upon CLIP-C-RN by **7.0**% and surpasses the second-best CLIP-based method, TPT-RN, by **3.8**%. Among the "-B32" models, CausalDA-C-B32 excels with a margin of at least **10.5**% over previous methods.

Table 8: SF-OODG results (%) on **ImageNet variants**. L, ViL means label-required, ViL model-required, respectively.

| Method | Venue | L | ViL | ImageNet→X | | | | |
| --- | --- | --- | --- | --- | --- | --- | --- | --- |
| | | | | IN-V2 | IN-K | IN-A | IN-R | Avg. |
| Source | – | – | – | 62.7 | 22.2 | 0.7 | 35.1 | 30.2 |
| CLIP-RN | ICML21 | – | ✓ | 51.5 | 33.3 | 21.8 | 56.1 | 40.7 |
| CLIP-B32 | ICML21 | – | ✓ | 54.8 | 40.8 | 29.5 | 66.2 | 47.8 |
| CoOP-RN | IJCV21 | ✓ | ✓ | 55.4 | 34.7 | 23.1 | 56.6 | 42.4 |
| CoOP-B32 | IJCV21 | ✓ | ✓ | 58.2 | 41.5 | 31.3 | 65.8 | 49.2 |
| CoCoOp-RN | CVPR22 | ✓ | ✓ | 55.7 | 34.5 | 23.3 | 57.7 | 42.8 |
| CoCoOp-B32 | CVPR22 | ✓ | ✓ | 56.6 | 40.7 | 30.3 | 64.1 | 47.9 |
| TPT-RN | NeurIPS22 | ✗ | ✓ | 54.7 | 35.1 | 26.7 | 59.1 | 43.9 |
| ProGrad-RN | ICCV24 | ✓ | ✓ | 54.7 | 34.4 | 23.1 | 56.8 | 42.2 |
| ProGrad-B32 | ICCV24 | ✓ | ✓ | 57.4 | 41.7 | **31.9** | 66.5 | 49.4 |
| SHOT | ICML20 | ✗ | ✗ | 62.5 | 38.4 | 2.1 | 42.7 | 36.4 |
| GKD | IROS21 | ✗ | ✗ | 62.3 | 38.2 | 2.1 | 42.2 | 36.2 |
| NRC | NeurIPS21 | ✗ | ✗ | 61.5 | 0.9 | 1.3 | 28.9 | 23.1 |
| AdaCon | CVPR22 | ✗ | ✗ | 50.9 | 18.5 | 2.0 | 38.6 | 27.5 |
| CoWA | ICML22 | ✗ | ✗ | 62.7 | 37.8 | 1.7 | 46.3 | 37.1 |
| PLUE | CVPR23 | ✗ | ✗ | 53.0 | 18.5 | 2.0 | 38.6 | 28.0 |
| TPDS | IJCV24 | ✗ | ✗ | 62.9 | 35.0 | 2.9 | 48.4 | 37.3 |
| SHOT-B32 | ICML20 | ✗ | ✓ | **69.4** | 43.6 | 1.6 | 58.2 | 43.2 |
| GKD-B32 | IROS21 | ✗ | ✓ | 53.5 | 45.2 | 1.0 | 58.3 | 39.5 |
| NRC-B32 | NeurIPS21 | ✗ | ✓ | 66.2 | 18.6 | 15.4 | 38.0 | 34.6 |
| TPDS-B32 | IJCV24 | ✗ | ✓ | 69.0 | 38.9 | 0.6 | 58.7 | 41.8 |
| **CausalDA**-C-RN | – | ✗ | ✓ | 64.4 | 42.6 | 22.3 | 61.5 | 47.7 |
| **CausalDA**-C-B32 | – | ✗ | ✓ | 64.7 | **48.4** | 30.6 | **71.0** | **53.7** |

## 4.7 Further analysis

All experiments use the strong variant CausalDA-C-B32; the suffix "-C-B32" is omitted hereafter.

### 4.7.1 Causal vs. statistical SFDA

In this strategic evaluation, we contrast CausalDA with two state-of-the-art statistical SFDA methods, DIFO (Tang et al., 2024c) and ProDe (Tang et al., 2025), both of which are equipped with the same ViL model. Our analysis focuses on two critical modeling aspects, examined across various SFDA settings: *Domain fitness*: Evaluated under Closed-set conditions; *Domain generalization*: Assessed across Generalized, Partial-set, Open-set, and SF-OODG settings.

Table 9: Causal vs. statistical SFDA. OH/VDA/DN126: **Office-Home/VisDA/DomainNet-126**. S/T: Source/Target domain; $H$: Harmonic mean accuracy. $H_g$: Average over all settings except closed-set.

| Method | Venue | Closed-set | | | | Generalized | | | Open-set | Partial-set | SF-OODG | | | | | |
|---|---|---|---|---|---|---|---|---|---|---|---|---|---|---|---|---|
| | | OH | VDA | DN126 | $Avg.$ | S | T | $H$ | T | T | IN-V2 | IN-K | IN-A | IN-R | $Avg.$ | $H_g$ |
| Source | – | 59.2 | 49.2 | 54.7 | 54.4 | 98.1 | 59.2 | 73.8 | 46.6 | 62.8 | 62.7 | 22.2 | 0.7 | 35.1 | 30.2 | 53.4 |
| DIFO | CVPR24 | 83.1 | 90.3 | 80.0 | 84.5 | 78.0 | 83.1 | 80.5 | 75.9 | 85.6 | 59.6 | 37.7 | 25.2 | 66.4 | 47.2 | 72.3 |
| ProDe | ICLR25 | **84.5** | **91.0** | **85.0** | **86.8** | 85.1 | 84.5 | 84.8(±0.2) | 82.6 | 84.2 | 62.1 | 45.8 | 24.7 | 69.7 | 50.6 | 75.5 |
| **CausalDA** | – | 84.3 | 90.3 | 82.2 | 85.6 | **86.1** | 84.3 | **85.3(±0.2)** | **84.0** | **87.1** | **64.7** | **48.4** | **30.6** | **71.0** | **53.7** | **77.5** |

Table 10: Unification comparison results (%). $H_{all}$: Average over all settings; $H_{wrg}$: The largest percentage shortfall of a method from the per-setting best across all settings; $H_{loso}$: Omit one setting at a time and average the scores over the rest.

| Method | $H_{wrg}(\downarrow)$ | $H_{loso}(\uparrow)$ | | | | | $H_{all}(\uparrow)$ |
|---|---|---|---|---|---|---|---|
| | | w/o Closed-set | w/o Generalized setting | w/o Open-set | w/o Partial-set | w/o SF-OODG | |
| DIFO | 12.1 | 72.3 | 73.3 | 74.4 | 72.0 | 81.6 | 74.7 |
| ProDe | 5.77 | 75.5 | 76.0 | 76.6 | 76.2 | 84.6 | 77.7 |
| **CausalDA** | **1.38** | **77.5** | **77.6** | **77.9** | **77.1** | **85.5** | **79.1** |

Based on this evaluation, we draw the following key observations: (1) Domain fitness (Closed-set SFDA): As presented in Tab. 9, CausalDA is indeed outperformed by both DIFO and ProDe in the Closed-set setting. This is attributed to the inherent statistical learning nature of these alternatives, which tends to overfit their models towards the specific target domain. (2) Under the Generalized SFDA setting, CausalDA demonstrates superior performance in H-accuracy, also exhibiting the least drop in source domain accuracy (see S-accuracy). This is because the captured causality mitigates the risk of forgetting the source domain. (3) Except for the Closed-set scenario, CausalDA consistently stands out with strong performance on the other four settings. This highlights the superior domain generalization capabilities achieved through our method's causal knowledge discovery. Importantly, our approach provides the best overall solution across all these diverse settings. Collectively, these results validate that our approach strikes the optimal balance between domain fitness and generalization, thereby laying a solid foundation to serve as a unified SFDA solution.

**Unification comparison.** To underscore CausalDA's advantage over DIFO and ProDe, we perform a unification evaluation using the metrics in Eq. (17). The comparison in Tab. 10 shows that CausalDA attains the highest score in all scenarios. This indicates that CausalDA delivers robust, stable performance across diverse settings. Moreover, it exhibits no apparent weakness in any single scenario (cf. $H_{wrg}$), and its all-round superiority is not driven by any individual setting (cf. $H_{loso}$). Together, this evidence supports our claim of being towards a universally adaptable framework.

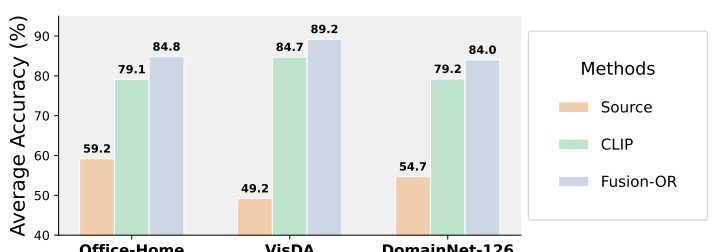

Figure 6: Zero-shot fusion results (Fusion-OR) of CLIP and the source model (Source) under the Closed-set SFDA setting on the **Office-Home**, **VisDA**, and **DomainNet-126** datasets. This fusion method follows the logical OR principle: An input instance is considered correctly classified if either CLIP or the source model predicts its ground truth category.

### 4.7.2 Analysis for invariant factors discovery

We conduct this analysis in two aspects: (1) complementarity of internal and external invariant factors, and (2) analysis for the discovery process based on pseudo-label.

**Complementarity of internal and external invariant factors.** As aforementioned, invariant factors originate from both the domain data itself (encoded internal component) and external knowledge sources

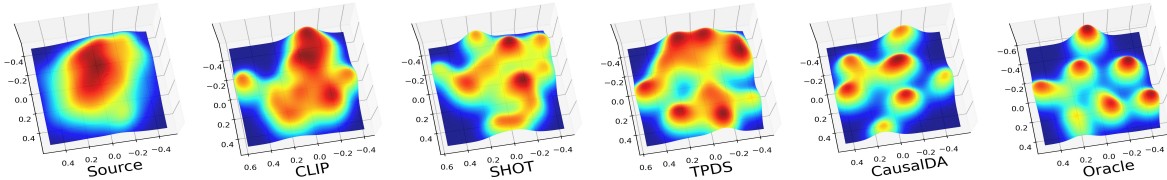

Figure 8: Logit distribution visualization on transfer task Ar→Cl in **Office-Home**. Oracle: Model trained on the real labels of domain Cl. For a clear view, the first ten categories are displayed.

(encoded external component). To illustrate this complementarity, we conduct an intuitive empirical experiment on the target domain. Specifically, we fuse the zero-shot predictions of the source model (Source) and a ViL model like CLIP using a logical OR operation. That is, if either CLIP or Source correctly predicts the true category of an input instance $x$, we consider $x$ correctly classified. As shown in Fig. 6, Fusion-OR significantly outperforms both Source and CLIP individually, reinforcing the plausibility of our internal-external partition.

**Pseudo-label-based analysis for discovery process of invariant factors**. In CausalDA, the external invariant elements $D_e$ are identified and converted into pseudo-labels. Thus, the quality of these pseudo-labels serves as an indicator of the discovery process. To evaluate this, we compare pseudo-label-based methods SHOT, COWA, and GKD, along with CLIP's zero-shot predictions (denoted as CLIP). The results, shown in Fig. 7, demonstrate that CausalDA-PL achieves significantly higher pseudo-label accuracy than prior methods, benefiting from CLIP-assisted $S_e$ discovery. Furthermore, the performance curve of CausalDA remains consistently above that of CausalDA-PL, with a significant gap throughout training. This highlights the effectiveness of capturing internal invariant elements $D_i$, further enhancing model performance.

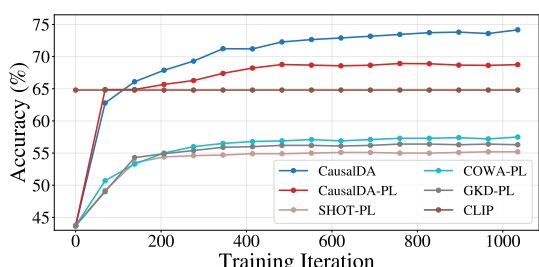

Figure 7: Dynamics of pseudo-label accuracy on task Ar→Cl in the **Office-home** dataset. The method "X-PL" means the corresponding accuracy of pseudo-label generated in X.

### 4.7.3 Model analysis

**Feature distribution visualization**. Feature distribution provides an intuitive way to evaluate classification performance. We compare with Source (source model), CLIP, SHOT, TPDS, and Oracle (trained on domain Cl with real labels). Fig. 8 presents the logit visualization results using a 3D density chart. As we move from Source to CausalDA, the clustering effect progressively improves, with CausalDA exhibiting the most structured distribution, closely resembling Oracle.

**Ablation study**. In CausalDA, we separately discover the invariant factors $\{D_e, D_i\}$. To isolate the effect of this strategy, we evaluate four variations: (i) CausalDA-w-$D_i$: We use only $\mathcal{L}_{UN}$ from Eq. (16) as the objective for model training, allowing us to assess the impact of capturing $D_i$. (ii) CausalDA-w-$D_e$: We apply pseudo-label supervised learning, regulated by $\mathcal{L}_{SCE}$, where the pseudo-labels are generated from $D_e$ discovery, regulated by $L_{EC}$. This isolates the effect of capturing $D_e$. (iii) CausalDA-w-CLIP: We directly use CLIP's zero-shot results as pseudo-labels to regulate the discovery of $D_i$. (iv) CausalDA-w-P1: We retain only the phase-1 objective, removing the loss $\mathcal{L}_{IC}$ from Eq. (16).

Table 11: Ablation study results (%). CS, OS, PS, and GS mean the Closed-set, Open-set, Partial-set, and Generalized SFDA settings, respectively.

| Method | Office-Home | | | | | IN-K |
|---|---|---|---|---|---|---|
| | CS | OS | PS | GS | Avg. | SF-OODG |
| Source | 59.2 | 46.6 | 62.8 | 73.1 | 60.4 | 30.2 |
| **CausalDA**-w-$D_i$ | 71.3 | 72.1 | 71.6 | 76.6 | 72.9 | 36.9 |
| **CausalDA**-w-$D_e$ | 82.5 | 81.4 | 85.5 | 84.9 | 83.6 | 44.2 |
| **CausalDA**-w-CLIP | 82.0 | 78.8 | 85.1 | 83.3 | 82.3 | 44.2 |
| **CausalDA**-w-P1 | 80.3 | 38.1 | 80.7 | 78.1 | 69.3 | 40.8 |
| **CausalDA** | **84.3** | **84.0** | **87.1** | **85.2** | **85.1** | **48.4** |

Tab. 11 presents the results of these methods across four SFDA settings and SF-OODG. Specifically, both CausalDA-w-$D_i$ and CausalDA-w-$D_e$ outperform the Source model, but fall significantly behind the full CausalDA version. This suggests that capturing either $D_e$ or $D_i$ independently is effective, but both are necessary for optimal performance. The comparison between CausalDA-w-CLIP and CausalDA highlights the importance of $L_{EC}$-based prompt learning for $D_e$ discovery. Finally, CausalDA-w-P1 performs considerably worse than CausalDA, emphasizing the value of our two-stage design in the invariant factor capture pipeline.

**Scalability study of IC instantiation**. The objective $\mathcal{L}_{IC}$ in Eq. (16), which regulates $D_i$ discovery, is a component with multiple implementation options. To evaluate its scalability, we present three alternative instantiations for the two regulators in $\mathcal{L}_{IC}$. For $\mathcal{L}_{UN}$, we consider the following alternatives: (i) CausalDA+ENT: Entropy regularization without category balance constraints, (ii) CausalDA+NRC: Data local structure-based regularization, (iii) CausalDA+TPDS: Domain shift control-based regularization. For $\mathcal{L}_{SCE}$, we test three popular methods: 1-Norm ($||\cdot||_1$), 2-Norm ($||\cdot||_2$), and KL Divergence.

As shown in Tab. 12, in the Closed-set setting, CausalDA+TPDS achieves the best results, with other methods lagging by up to **1.9**% (on VisDA). This indicates that these typical regulators do not significantly affect model performance in this setting. In the SF-OODG setting, all methods (except CausalDA+KL) show a slight accuracy drop compared to CausalDA. For CausalDA+ENT, removing the category balance regularization leads to a performance decrease. The performance loss for CausalDA+NRC and CausalDA+TPDS is attributed to the fact that these methods do not specifically address the SF-OODG setting.

Table 12: Scalability study (%) of IC instantiation.

| Loss | Method | Closed-set | | SF-OODG |
|---|---|---|---|---|
| | | **Office-Home** | **VisDA** | **IN-K** |
| $\mathcal{L}_{UN}$ | **CausalDA**+ENT | 83.5 | 90.4 | 46.8 |
| | **CausalDA**+NRC | 83.9 | 89.0 | 47.5 |
| | **CausalDA**+TPDS | **84.6** | **90.9** | 45.0 |
| $\mathcal{L}_{SCE}$ | **CausalDA**+$||\cdot||_1$ | 83.4 | 89.3 | 44.9 |
| | **CausalDA**+$||\cdot||_2$ | 83.8 | 89.2 | 40.3 |
| | **CausalDA**+KL | 84.5 | 90.0 | **49.1** |
| | **CausalDA** | 84.3 | 90.3 | 48.4 |

For $\mathcal{L}_{SCE}$, the results show that probability distribution approximation (KL Divergence) outperforms feature-based methods ($||\cdot||_1$ and $||\cdot||_2$) in preserving model performance in the presence of external invariant factors.

**Effect of prompt initialization**. In CausalDA, we employ prompt learning to encode the external invariant factors $D_e$. Here, we examine four different prompt initialization strategies (Tab. 13): (i) the conventional constant strategy (rows 1–2), (ii) the sentence strategy (rows 3–4), (iii) the uncertainty strategy (rows 5–7), and (iv) an innovative approach of sentence initialization with different phrase (rows 8–9), which considers the lack of precise supervision in SFDA.

Table 13: Prompt initialization study (%).

| # | Initialization template | Closed-set | | SF-OODG |
|---|---|---|---|---|
| | | **Office-Home** | **VisDA** | **IN-K** |
| 1 | 'X [CLASS].' (#X=4) | 83.6 | 90.1 | 48.4 |
| 2 | 'X [CLASS].' (#X=16) | 82.5 | 89.6 | 46.1 |
| 3 | 'There is a [CLASS].' | 84.6 | 90.2 | 48.6 |
| 4 | 'This is a photo of a [CLASS].' | 84.7 | 89.9 | **48.8** |
| 5 | 'This is maybe a photo of a [CLASS].' | 83.9 | 90.2 | **48.8** |
| 6 | 'This is almost a photo of a [CLASS].' | **84.8** | **90.3** | 48.4 |
| 7 | 'This is definitely a photo of a [CLASS].' | 84.5 | 90.0 | 48.1 |
| 8 | 'a picture of a [CLASS].' | 84.6 | 89.8 | 48.6 |
| 9 | 'a photo of a [CLASS].' | 84.3 | **90.3** | 48.4 |

Tab. 13 shows the comparison results for these nine templates across both the Closed-set setting and SF-OODG. The results indicate that the different templates do not cause significant performance differences, suggesting that CausalDA is relatively insensitive to prompt selection. However, when comparing the constant strategy to the others, it becomes clear

Table 14: Training resource demands on Ar→Cl in Office-Home. The cost variance is measured against SHOT.

| Item / Method | SHOT | DIFO | CausalDA |
|---|---|---|---|
| GPU memory cost (GB) | 8.7 | 10.1 (↑1.4) | 10.8 (↑2.1) |
| Training times (s) | 424 | 879 (↑455) | 755 (↑331) |

that initializing with semantic information is the more effective option, which aligns with our expectations.

**Training resource demands.** In this part, we compare training resources on the transfer task Ar→Cl in Office-Home. For comparison, we select previous SOTA methods SHOT and DIFO. Unlike SHOT, which does not require knowledge of the ViL model, DIFO leverages this external knowledge, aligning with our method. The results presented in Tab. 14 indicate that our CausalDA, alongside DIFO, requires more computational resources and training time compared to SHOT. This is expected, as SHOT does not incur the additional overhead required to transfer task-specific knowledge from a pretrained ViL model. In the ViL model-based group, our CausalDA does not impose significant additional training costs, requiring a computational resource level similar to DIFO. This demonstrates that our performance gains stem fundamentally from the methodological design, rather than simply trading computational resources for accuracy.

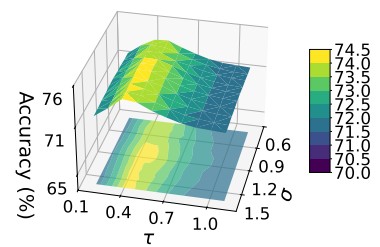

Figure 9: Parameter sensitiveness study $(\sigma, \tau)$ on transfer task Ar→Cl in **Office-Home**.

**Parameters sensitiveness**. Taking the transfer task Ar→Cl in Office-Home for example, we evaluate the sensitivity of CausalDA by varying the hyper-parameters $0.2 \leq \sigma \leq 1.1$, $0.7 \leq \tau \leq 1.6$ at the step of 0.1. Fig. 9 indicates that both corresponding terms matter, whilst their sensitivity to performance is not high.

## 5 Conclusion

We introduce a more challenging Unified SFDA problem that comprehensively addresses all specific scenarios within a unified framework. To tackle this, we propose CausalDA, a novel causality-inspired approach, fundamentally differing from previous methods based on statistical dependence. Specifically, we design a self-supervised invariant information maximization to capture external invariant elements, convert them into pseudo-labels, and use them to guide the discovery of internal invariant factors. Building on our theoretical insights, we implement this invariant information maximization using a network-based variational mutual information approach. Extensive experiments across various SFDA settings and SF-OODG validate the superiority of CausalDA over existing methods.

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

## A   A Proof of Lemma 1

**Restatement of Lemma 1** *Given random variables $Z_1$, $X_1$, $Y_1$ where $X_1$, $Y_1$ satisfy a mapping $f_1 : X_1 \mapsto Y_1$. When $f_1$ is compressed, i.e., the output's dimension is smaller than the input's,*

$$I(Z_1, X_1) \geq I(Z_1, Y_1) \tag{18}$$

**Proof.** Since mapping $f_1$ is compressed, the entropy of $X_1$, $Y_1$ have $H(X_1) \geq H(Y_1)$. As for $I(Z_1, X_1)$, it has

$$
\begin{aligned}
I(Z_1, X_1) &\geq I(Z_1, X_1) - H(Z_1|Y_1) - I(X_1, Y_1) \\
&\geq [H(Z_1) + H(X_1) - H(Z_1, X_1)] - H(Z_1|Y_1) \\
&\quad - [H(X_1) + H(Y_1) - H(X_1, Y_1)] \\
&= H(Z_1) - H(Z_1|Y_1) - H(Y_1) + H(X_1, Y_1) \\
&\geq I(Z_1, Y_1) - H(Y_1) + \max(H(X_1), H(Y_1)) \\
&\geq I(Z_1, Y_1) - H(Y_1) + H(X_1) \\
&\geq I(Z_1, Y_1) - H(Y_1) + H(Y_1) \\
&= I(Z_1, Y_1)
\end{aligned}
\tag{19}
$$

## B  A Proof of Theorem 1

**Lemma 2** *(Data Processing Inequality, DPI) (Cover & Thomas, 2006) Given random variables $X$, $Y$ and $Z$. If there is a Markov chain structure $X \rightarrow Y \rightarrow Z$, then $I(X,Y) \geq I(X,Z)$, which suggests that the mutual information between the prediction $X$ and the post-processed pseudo-label $Z$ is strictly bounded by the mutual information between $X$ and the original feature $Y$.*

**Restatement of Theorem 1** *Suppose that there are five random variables $Z$, $V$, $Z'$, $Y$ and $Y'$. Among them, $Z$ represents the target domain knowledge; $V$, $Z'$ and $Y$ express the input prompt, intermediate features of the ViL model, and predictions, respectively; $Y'$ depicts a pseudo-label that has a learnable functional relationship $f_w$ ($w$ is parameters) with $Z'$, i.e., $Y' = f_w(Z')$. For the mutual information maximization objective, we have the following lower bound:*

$$\max_{\boldsymbol{v}_d} I\left(Z, Z'\right) - I\left(Z', Y\right) \geq \max_{\boldsymbol{v}_d, w} \underbrace{I\left(Z, Z'\right)}_{T_1} + \underbrace{I\left(Y', Y\right), Y' = f_w\left(Z'\right)}_{T_2}. \tag{20}$$

**Proof.** In our network architecture, both the model prediction $Y$ and the pseudo-label $Y'$ are derived exclusively from the ViL's intermediate feature $Z'$. Specifically, $Y = \phi(Z')$ and $Y' = f_w(Z')$. Because neither $Y$ nor $Y'$ incorporates any additional external information other than what is provided by $Z'$, they are conditionally independent given $Z'$, namely:

$$P(Y, Y'|Z') = P(Y|Z')P(Y'|Z') \tag{21}$$

According to the information theory, this conditional independence expressed in Eq. (21) strictly defines the Markov chain $Y \rightarrow Z' \rightarrow Y'$ (Cover & Thomas, 2006). By applying DPI Lemma 2 (above) to this Markov chain, we have

$$I(Y', Y) \leq I(Z', Y) \tag{22}$$

By adding the mutual information term $I(Z, Z')$ to both sides of the inequality, we obtain:

$$I(Z, Z') + I(Y', Y) \leq I(Z, Z') + I(Z', Y) \tag{23}$$

By taking the maximum over the prompt context $v_e$ and the network parameters $w$, we establish the following tractable lower bound for our original causal discovery objective, with $Y' = f_w(Z')$.

$$\max_{v_e} \left[I(Z, Z') + I(Z', Y)\right] \geq \max_{v_e, w} \left[I(Z, Z') + I(Y', Y)\right]. \tag{24}$$

