# OpenReview forum: "Unified Source-Free Domain Adaptation via Causality-Inspired Latent Invariant Factor Learning"
_TMLR — Under review for TMLR_

### Review · Reviewer_LL7a · 2026-05-17

**Summary Of Contributions:**

This paper studies source-free domain adaptation (SFDA) when the relationship between source and target label spaces is unknown. It proposes a broader Unified SFDA setting that covers closed-set, open-set, partial-set, generalized SFDA, and OOD-style evaluation.

The method, CausalDA, uses CLIP as an external knowledge source. It learns prompt-based external factors from CLIP and combines them with internal factors learned from the source model and target data. The method is motivated by causal representation learning and uses a self-supervised information bottleneck objective.

The empirical results are strong across several SFDA settings.

Pros.

1. The paper addresses a realistic limitation of SFDA. In practice, the target label space is often unknown. A unified setting is therefore useful. The method is tested across multiple settings: closed-set, generalized, open-set, partial-set, and SF-OODG. This makes the empirical evaluation more convincing than a single-setting study.

3. Using CLIP as external semantic knowledge is a reasonable design choice. It is especially useful when source data and target labels are unavailable.

4. The split between external factors from CLIP and internal factors from the adapted target model is intuitive. The ablations suggest both parts contribute.

5. CausalDA performs competitively across several datasets and settings. The unified evaluation also suggests good cross-setting stability. The paper studies variants using only internal factors, only external factors, CLIP pseudo-labels, and phase-1-only training. These experiments help justify the design choices.

**Audience:**

Yes

**Audience Explanation:**

Domain adaptation is an important problem in the field of machine learning.

**Claims And Evidence:**

No

**Claims Explanation:**

I list the claims that are not well-supported by the evidence below.

**Claim 1: CausalDA discovers latent causal factors; Justification: Weak.**

The paper does not show causal identifiability, true interventions, or recovery of ground-truth causal factors.

Assessment: Not fully supported. Better phrased as “causality-inspired latent factor learning.”

**Claim 2: Logit space is suitable for causal analysis; Justification: Partial.**

Logits are compact and semantic. But calling them “post-intervened” is not rigorously justified.

Assessment: Partially supported.

**Claim 3: CLIP provides external causal factors; Justification: Partial.**

CLIP provides useful semantic knowledge. But CLIP knowledge is not necessarily causal.

Assessment: Supported as external semantic prior, not as causal factor evidence.

**Claim 4: The information bottleneck has theoretical guarantees; Justification: Limited.**

The theory supports the optimization surrogate. It does not prove causal recovery.

Assessment: Supported only in a narrow optimization sense.

The above claims should be toned down. The paper should avoid saying that CausalDA discovers causal factors or reveals causal mechanisms, because the current theory and experiments do not establish causal identifiability or causal recovery. The claim that CLIP provides external causal factors should also be weakened to CLIP providing an external semantic prior. Similarly, the statement that the logit space is post-intervened should be phrased more cautiously, since domain-shifted logits are not equivalent to a formal causal intervention. The “theoretical guarantees” should be described as guarantees for the proposed information-bottleneck surrogate, not for causal discovery.

Overall, the causal language should be changed from “causal discovery” to “causality-inspired invariant/semantic factor learning.”

**Requested Changes:**

Please see the claim part above.

---

> ### Author Response · Authors · 2026-06-11
> **Response to Reviewer LL7a**
>
> Dear reviewer **LL7a**, we are grateful for your time in reviewing this paper and we provide a point-to-point response to your comments below.
>
> **Q1**. **Claim 1: CausalDA discovers latent causal factors; Justification: Weak.** The paper does not show causal identifiability, true interventions, or recovery of ground-truth causal factors. Assessment: Not fully supported. Better phrased as “causality-inspired latent factor learning.”
>
> **R:** Appreciate the insightful comment. We agree with the reviewer's concern that the latent variables captured by CausalDA might not strictly equate to formal causal factors. Instead, they are more rigorously characterized as latent invariant factors, which can be viewed as a practical proxy and empirical manifestation of underlying causal structures under the domain shifts. This choice leverages the shared cross-domain insensitivity of invariant factors and causality.
>
> To address this concern, we have updated our manuscript in two aspects：
>
> * We have revised the entire manuscript to **reframe our framework as "causality-inspired latent invariant factor learning"**, replacing over-claimed causal jargon with this more precise descriptor.
> * We have added a **Remark** at the end of Section 3.3 to explicitly clarify that the proposed causality principle (Eq. (3)) is a conceptual hypothesis, which inspires the CausalDA method to focus on learning latent invariant factors.
>
>
> **Q2**. **Claim 2: Logit space is suitable for causal analysis; Justification: Partial.** Logits are compact and semantic. But calling them “post-intervened” is not rigorously justified. Assessment: Partially supported.
>
> **R:** Great comment. In this work, we do not claim that the logit space provides a fully rigorous mathematical justification for the causal analysis. Instead, from the perspective of the cited literature, viewing the logit space as post-intervened provides a plausible and intuitive explanation for its effectiveness. Admittedly, not all empirical strategies or methodological choices in deep learning can be strictly proven from a mathematical standpoint.
>
> Accordingly, **at the end of the first paragraph in Sec. 3.3**, we have tempered our claims regarding the 'post-intervened' nature of the logit space, explicitly framing it as a reasonable interpretation grounded in existing literature rather than an absolute claim.
>
>
> **Q3**. **Claim 3: CLIP provides external causal factors; Justification: Partial.** CLIP provides useful semantic knowledge. But CLIP knowledge is not necessarily causal. Assessment: Supported as external semantic prior, not as causal factor evidence.
>
> **R:** Thank you very much for pointing this out. The reviewer is correct that CLIP's knowledge is not necessarily causal. In this work, we do not treat CLIP as direct causal evidence, but rather as an external semantic prior that provides the necessary context to help uncover these external invariant factors, which are nearly intractable to detect when limited only to target domain observations and source model.
>
> We have updated the definitions of the internal and external causality provided in **the fourth paragraph of Sec. 3.3**.
>
>
> **Q4**. **Claim 4: The information bottleneck has theoretical guarantees; Justification: Limited.** The theory supports the optimization surrogate. It does not prove causal recovery. Assessment: Supported only in a narrow optimization sense.
>
> **R:** Geat comment. As indicated in our response to **Q1**, we have updated our theoretical formulation to **learning latent invariant factors**. This transition resolves the concern regarding whether the theoretical results guarantee causal recovery.

---

### Review · Reviewer_XwGQ · 2026-05-27

**Summary Of Contributions:**

The paper introduces unified SFDA, where a single method handles closed-set, open-set, partial-set, generalized SFDA, etc, without knowing the regime in advance. This method, CausalDA, takes a causal view by constructing an SCM in logit space and decomposing causal factors into external factors from frozen CLIP and internal factors learned from source-target adaptation. Training alternates between discovering external factors via a self-supervised information bottleneck to generate pseudo-labels, and learning internal factors using information maximization plus pseudo-label cross-entropy. The paper reports results across Office-Home, VisDA, DomainNet-126, and ImageNet variants.

**Audience:**

Yes

**Audience Explanation:**

The proposed framework is genuinely interesting and will be useful to the audience working on Domain adaptation and Domain Generalization

**Claims And Evidence:**

No

**Claims Explanation:**

**Strengths**

1) The problem formulation is interesting. The observation that deployed systems rarely know the target category relationship is correct and overlooked in the literature, and packaging the scattered SFDA settings into one framework is a useful contribution in itself.

2) The empirical coverage is broad. The ablations are reasonably thorough: the Fusion-OR experiment (Fig. 6) is a clean, intuitive motivation for the internal/external split, infact this could have been the main figure of the paper.  The pseudo-label dynamics plot (Fig. 7) isolates where the gains come from, and the prompt-initialization  sensitivity studies are useful

3) The design choices are intuitive, keeping CLIP frozen and adapting only a prompt avoids catastrophic forgetting, and operating in the compact logit space is reasonable.

**Weakness**

1) *The "causality" framing is loosely connected to the actual method*

The paper's identity rests on being causal rather than statistical, but the machinery that does the work is largely statistical.
Phase 1 is essentially CoOp-style prompt learning on CLIP, wrapped in an information bottleneck objective. Phase 2's L_UN (Eq. 15) is exactly the information-maximization loss (entropy minimization + KL-to-uniform), which is the core of SHOT. L_SCE is a distillation from CLIP pseudo-labels. So phase 2, the part labeled "internal causality discovery," is in practice SHOT + CLIP distillation.  The claim that the logit space is "post-intervened" is asserted, not demonstrated, the dis() operator in Eq. (2) is not define. The factorization in Eq. (3), writing P(y|S) as a product of two posteriors, is motivated by ICM, as per reviewers' understanding ICM concerns the independence of mechanisms, not the factorization of a posterior into a product, and the product of two conditional probabilities is not itself a normalized probability. The step needs justification.

2) *Inconsistencies in the loss function derivation*

The method's stated novelty rests on a self-supervised information bottleneck with theoretical guarantees (Sec.3.4, Theorem1). The objective (Eq.7) is to minimize $I(Z,Z') - I(Z',Y)$, and Theorem~1 provides an upper bound

$I(Z,Z') - I(Z',Y)   \le  T_1 - T_2$

$T_1 = I(Z,Z') $ and  $T_2 = I(Y’,Y)$

Minimizing this surrogate requires driving $T_1$  down and $T_2$  up. The implementation does not do this for $T_1$. To minimize a mutual-information term one needs an upper bound on it (e.g., a KL-to-variational-marginal bound, as in deep variational IB). What the paper invokes (Eq.9) is the Barber-Agakov bound, which is a lower bound on $I(Z,Z’)$, and is, by construction, the standard tool for MI maximization . $L_{\mathrm{VMI}}$ (Eq.10) is this lower bound, and Eq.(13), $\min \big(L_{\mathrm{PMI}} - \alpha L_{\mathrm{VMI}}\big)$ with $\alpha>0$, maximizes it. So the compression term is handled with the wrong type of bound and then optimized in the wrong direction; the two do not cancel into a valid minimization-they compound, pushing $I(Z,Z')$ away from the
minimum the theorem requires.

Crucially, a lower bound carries no order relation to the objective it is meant to control. We have $a \le b$ (Eq.8) with $b = T_1 - T_2$, and the optimized quantity satisfies $c \le T_1$ (the Barber-Agakov bound, Eq.9). Since $T_1$ enters $b$ on its upper side while $c$ lies below $T_1$, both $a$ and $c$ sit on the lower side of a common reference; minimizing-or, as here, maximizing-$c$
therefore transmits no guarantee to $a$.

In the proof of Lemma 1 in the appendix, please check the derivation, it seems the authors have dropped the term H(Z1, Y1). The conclusion of the lemma still holds, the authors can appeal to the standard data processing inequality and it does not require any compression assumption it just needs Markov chain X1->Y1->Z1.

*On the empirical Side:*

Please consider reporting the standard deviation where the gain is sub 1\%.

**Requested Changes:**

Please fix the following

1) As detailed in the weaknesses, the method as implemented is largely standard (prompt learning + SHOT-style information maximization + CLIP distillation). Please either add evidence that engages the causal claim more directly (e.g., an interventional or counterfactual analysis), or change the framing to "causally-inspired" and adjust the contribution statements accordingly.

2)  Mathematical inconsistencies, Fix wherever it can be done, like Lemma 1 can be easily fixed by appealing to data processing inequality. The serious issue is the flow of logic from Eq.7 to Eq.13. The authors can do one of the two
a)  Provide the correct derivation.   b) If a correct derivation is not feasible, reframe the method as empirically motivated and remove the "information bottleneck with theoretical guarantees" claim, justifying the losses on intuition and ablation evidence instead. Option (b) is acceptable, but not in combination with retaining the theoretical-guarantee framing.

3) Authors can consider reporting standard deviations for the tables where the gain is sub 1\% atleast for the best and the second best data points.

---

> ### Author Response · Authors · 2026-06-11
> **Response to the Reviewer XwGQ**
>
> Dear reviewer **XwGQ**, we are grateful for your time in reviewing this paper and we provide a point-to-point response to your comments below.
>
> **A1**. As detailed in the weaknesses, the method as implemented is largely standard (prompt learning + SHOT-style information maximization + CLIP distillation). Please either add evidence that engages the causal claim more directly (e.g., an interventional or counterfactual analysis), or change the framing to "causally-inspired" and adjust the contribution statements accordingly.
>
> **R:** Appreciate the insightful comment. We have toned down the causality discovery-related claims from two main aspects:
>
> * We have revised the entire manuscript to **reframe our framework as "causality-inspired latent invariant factor learning"**, replacing over-claimed causal jargon with this more precise descriptor.
> * We have added a **Remark** at the end of Section 3.3 to explicitly clarify that the proposed causality principle (Eq. (3)) is a conceptual hypothesis, which inspires the CausalDA method to focus on learning latent invariant factors.
>
> The detailed reply can refer to the response to **Q1**, **Q2**, **Q3** and **Q4** of the reviewer **LL7a**.
>
> **A2**. Mathematical inconsistencies, Fix wherever it can be done, like Lemma 1 can be easily fixed by appealing to data processing inequality. The serious issue is the flow of logic from Eq.7 to Eq.13. The authors can do one of the two a) Provide the correct derivation. b) If a correct derivation is not feasible, reframe the method as empirically motivated and remove the "information bottleneck with theoretical guarantees" claim, justifying the losses on intuition and ablation evidence instead. Option (b) is acceptable, but not in combination with retaining the theoretical-guarantee framing.
>
>
> **R:** We sincerely thank the reviewer for pointing out this issue, and we apologize for the theoretical oversight in our initial draft.
>
> Guided by the reviewer's insightful feedback, we carefully re-examined our theoretical derivations and code implementation. We found that our CausalDA method essentially performs the following optimization:
>
> $${\max_{v_e} I ({Z,{Z{'}}}) + I ({{Z{'}},Y})}.$$
>
> Furthermore, we sincerely apologize for the theoretical flaw in our original proof of Theorem 1; as the reviewer rightly implied, the inequality $H(Z'|Y') \ge H(Z')$ violates basic information-theoretic principles (since conditioning reduces entropy). To rigorously address this, we have derived a revised Theorem 1 based on the Data Processing Inequality (DPI), which establishes the following tractable lower bound:
> $$
> {\max_{v_e} I ({Z,{Z{'}}}) + I ({{Z{'}},Y})} \geq {\max_{v_e,w} I(Z,Z{'})} + I(Y{'},Y), Y{'} = f_{w} (Z{'}).
> $$
> In this lower bound, the first term aligns the generic external knowledge of the ViL model ($Z'$) with the target-domain representations ($Z$), ensuring that the extracted features $Z_e$ are specifically tailored to the target domain. The second term adopts a self-supervised formulation, where the pseudo-label $Y'$ is dynamically calibrated by an inverse-variance weight $1/\Sigma_Z$, with $\Sigma_Z$ derived from $I(Z', Z)$.
>
> Under the Gaussian assumption, $I(Z', Z)$ functions as a Mean-Square Error that inherently measures domain sensitivity. Thus, high-variance dimensions reflect severe data fluctuations, corresponding to non-causal noise ($U$, e.g., spurious backgrounds). Conversely, stable, low-variance dimensions encapsulate domain-invariant causal semantics ($S_e$, e.g., intrinsic shapes). Consequently, inverse-variance weighting structurally suppresses brittle domain artifacts while amplifying robust causal features. By utilizing $1/\Sigma_Z$ to parameterize the mapping $f_w$, we explicitly force the prediction mechanism to focus on these external, invariant causal factors.
>
> In the revised version, **we have rewritten Sec. 3.4.1 "External invariant factor discovery"** and the proof of revised Theorem 1 has been provided at the end of the paper's appendix.
>
>
> **A3**. Authors can consider reporting standard deviations for the tables where the gain is sub 1% atleast for the best and the second best data point.
>
>
> **R:** Great suggestion. We have added the standard deviations to the H-result of ProDe and CausalA in Table 9.

---

### Review · Reviewer_Vy5z · 2026-06-16

**Summary Of Contributions:**

This paper introduced a novel CausalDA method for unified source-free domain adaptation (SFDA). By discovering the causal relationship between latent invariant variables and model decisions, the proposed model could be applied to various SFDA settings. The internal-external causality decomposition sought to find complementary causal factors from both the external pretrained ViL model and the target model. Experimental results demonstrated that the proposed CausalDA method consistently achieved better performance than baselines.

**Audience:**

Yes

**Audience Explanation:**

The proposed method provides a flexible way to handle various source-free domain adaptation problems in real-world applications.

**Claims And Evidence:**

Yes

**Claims Explanation:**

Most claims are well supported and validated in theoretical analysis and empirical results. However, there are some concerns.

(1) It states that "Considering that the information from both domains is not necessarily complete, these proxy factors are disentangled into two complementary parts: (i) external invariant factors and (ii) internal invariant factors." The disentanglement between external and internal invariant factors can be justified.

(2) The factorization of Eq. (3) is hard to follow. What does each factorized term imply? Why does the conditional independence assumption support this factorization?

**Requested Changes:**

(1) The "semantic shift" in Table 1 can be explained with examples.

(2) Table 2 shows that "using logits for intervention" for the CausalDA method. It is good to explain how it works.

(3) The implementation details in Algorithm 1 can be provided, especially for optimizing $\mathcal{L}\_{EC}$ and $\mathcal{L}\_{IC}$ iteratively.

(4) The running efficiency of CausalDA can be analyzed compared to baselines.

---

> ### Author Response · Authors · 2026-06-18
> **Response to Reviwer Vy5z**
>
> Dear reviewer **Vy5z**, we are grateful for your time in reviewing this paper and we provide a point-to-point response to your comments below.
>
>
>
> **C1**. It states that "Considering that the information from both domains is not necessarily complete, these proxy factors are disentangled into two complementary parts: (i) external invariant factors and (ii) internal invariant factors." The disentanglement between external and internal invariant factors can be justified.
>
>
> **R:** Thanks for this comment. The partition of (i) external invariant factors and (ii) internal invariant factors stems from the internal-external causality decomposition within our proposed **latent causality hypothesis for unified SFDA (Sec. 3.3)**. As discussed in **the fourth paragraph of this section**, this decomposition is motivated by the fact that identifying all relevant causal relationships solely from the source and target domains is often infeasible due to limited training data and restricted domain-specific knowledge. To overcome this limitation, we introduce a pretrained ViL model, such as CLIP, as an external knowledge source to assist in discovering these causal relationships. Consequently, to clearly distinguish the origins of the causality, we explicitly partition it into internal and external components.
>
> Importantly, **as noted in the final sentence of that same paragraph**, the soundness of this division is empirically validated by our experimental results detailed in Sec. 4.2 and Sec. 4.7.2.
>
>
>
> **C2**. The factorization of Eq. (3) is hard to follow. What does each factorized term imply? Why does the conditional independence assumption support this factorization
>
> **R:** Great comment. Importantly, the proposed causality principle (Eq. (3)) is not a fully causal structure in a strict mathematical sense, but rather a conceptual hypothesis. This hypothesis motivates the CausalDA method to focus on learning latent invariant factors. To explicitly clarify this distinction, we have added a **Remark** at the end of Sec. 3.3.
>
>
> Furthermore, we would like to clarify that the conditional independence assumption is a general theory derived from classic literature, which we adapt to our SFDA task. This principle has found widespread success in many domains, for example, the Bayes-based approach. In our context, inheriting this assumption crucially decouples the entangled components, transforming an intractable optimization problem into manageable sub-problems via a divide-and-conquer strategy.
>
>
>
>
> **C3**. The "semantic shift" in Table 1 can be explained with examples.
>
>
> **R:** Thanks for this comment. The semantic shift refers to that the source and target domain have different categories.
>
> We have explained Covariate-shift and Semantic-shift in the caption of Table 1.
>
>
>
> **C4**. Table 2 shows that "using logits for intervention" for the CausalDA method. It is good to explain how it works.
>
> **R:** Great comment. In this work, we do not claim that the logit space provides a fully rigorous mathematical justification for the causal analysis. Instead, from the perspective of the cited literature, viewing the logit space as post-intervened provides a plausible and intuitive explanation for its effectiveness. Admittedly, not all empirical strategies or methodological choices in deep learning can be strictly proven from a mathematical standpoint.
>
>
> Accordingly, **at the end of the first paragraph in Sec. 3.3**, we have tempered our claims regarding the 'post-intervened' nature of the logit space, explicitly framing it as a reasonable interpretation grounded in existing literature rather than an absolute claim.
>
>
>
> **C5**. The implementation details in Algorithm 1 can be provided, especially for optimizing and iteratively.
>
> **R:** Great comment. For clarity and conciseness, Algorithm 1 outlines the core framework of the proposed training pipeline. Complete configuration and implementation details are fully documented in our source code, which is guaranteed to be publicly released upon the publication of this work.
>
> Please let us know if any parts require further clarification; we would be more than happy to provide additional details.
>
>
>
>
> **C6**. The running efficiency of CausalDA can be analyzed compared to baselines.
>
> **R:** Great suggestion. We have included an additional experiment to evaluate the training resource demands. As shown in the table, while leveraging ViL models naturally incurs more overhead than SHOT (as we expected), our CausalDA maintains a similar computational cost to the ViL-based baseline DIFO. This confirms that our performance improvements are driven by methodological design, not simply by scaling up resources.
>
> |Item/Method|SHOT|DIFO|CausalDA|
> |-|-|-|-|
> |GPU memory cost (GB)|8.7|10.1|10.8|
> |Training times (s)|424|879|755|
>
>
> We have included these experimental results in the revised verion (see the **Training resource demands** part at the end of sec. 4.7.3).

---

### Review · Reviewer_gxpe · 2026-06-17

**Summary Of Contributions:**

The paper proposes CausalDA, which tackles Unified Source-Free Domain Adaptation, where a model must adapt to an unlabeled target domain without access to source data and without knowing whether the target classes are the same, extra, missing, or partially overlapping with the source classes.

The method adapts a source-trained model by combining useful knowledge from CLIP with what the target model learns from unlabeled target data. CLIP provides broad semantic guidance, while the target model learns target-specific patterns. Together, they help the model focus on stable features that transfer across domains instead of relying on domain-specific shortcuts.

Extensive experiments across Office-Home, VisDA, DomainNet-126, and ImageNet variants are reported, showing competitive or superior performance to scenario-specific SFDA baselines, including CLIP-based variants.

Strengths:

1. The authors identify an important practical limitation of current SFDA systems: they often assume prior knowledge of how source and target classes are related. Unified SFDA addresses this by placing different SFDA scenarios under a single setting, making the problem closer to real deployment where the target domain structure may be unknown.
2. The method is conceptually simple, it uses a frozen CLIP model and first adapts the CLIP prompt using target-model logits, then uses the adapted CLIP predictions as pseudo-label guidance for the target model.
3. The paper shows strong quantitative results across several SFDA settings. CausalDA performs well not only in closed-set SFDA, but also in open-set, partial-set, generalized SFDA, and OOD generalization, supporting the claim that it is a broadly effective unified method.

Weaknesses:

1. The paper makes its causal contribution sound stronger than it really is. It says CausalDA discovers hidden causal factors, but it does not prove that these factors are truly causal. It does not perform real interventions or show that the learned factors match ground-truth causal variables. The results mainly show that the method learns features that are more stable across domains, which supports invariant learning but not actual causal discovery.
2. The method’s handling of unknown classes is not very clear. Although the paper claims to attempt to solve unified SFDA, CausalDA uses CLIP text prompts such as “a photo of a [CLS],” where [CLS] is a specific class name. This seems to require access to the candidate target class names. In realistic open-set or universal settings, the target domain may contain unknown classes whose names are not available in advance. The paper does not clearly explain how such classes are detected, rejected, or calibrated, which weakens the claim that the method fully handles unknown category relationships.

**Audience:**

Yes

**Audience Explanation:**

This is useful to researchers in the field of domain adaptation, distribution shift etc.

**Claims And Evidence:**

Yes

**Claims Explanation:**

The paper provides clear quantitative evidence that CausalDA performs well across several SFDA settings, including closed-set, generalized, open-set, partial-set, and OOD generalization. The unified comparison metrics also support the claim that the method is competitive across multiple settings.

**Requested Changes:**

Please refer to the weakness section

---

> ### Author Response · Authors · 2026-06-18
> **Response to Reviwer gxpe**
>
> Dear reviewer **gxpe**, we are grateful for your time in reviewing this paper and we provide a point-to-point response to your comments below.
>
>
>
>
> **B1**. The paper makes its causal contribution sound stronger than it really is. It says CausalDA discovers hidden causal factors, but it does not prove that these factors are truly causal. It does not perform real interventions or show that the learned factors match ground-truth causal variables. The results mainly show that the method learns features that are more stable across domains, which supports invariant learning but not actual causal discovery.
>
>
> **R:** Appreciate the insightful comment. We have toned down the causality discovery-related claims from two main aspects:
>
> * We have revised the entire manuscript to **reframe our framework as "causality-inspired latent invariant factor learning"**, replacing over-claimed causal jargon with this more precise descriptor.
> * We have added a **Remark** at the end of Section 3.3 to explicitly clarify that the proposed causality principle (Eq. (3)) is a conceptual hypothesis, which inspires the CausalDA method to focus on learning latent invariant factors.
>
>
> **B2**. The method’s handling of unknown classes is not very clear. Although the paper claims to attempt to solve unified SFDA, CausalDA uses CLIP text prompts such as “a photo of a [CLS],” where [CLS] is a specific class name. This seems to require access to the candidate target class names. In realistic open-set or universal settings, the target domain may contain unknown classes whose names are not available in advance. The paper does not clearly explain how such classes are detected, rejected, or calibrated, which weakens the claim that the method fully handles unknown category relationships.
>
>
> **R:** Thank you for the comment. In the open-set or universal setting, the prompt “a photo of a [CLS]” does not require access to unknown target class names. Here, [CLS] is instantiated only with source class names. For example, on Office-Home [1] open-set adaptation, the source domain contains 25 known classes, while the target domain contains 65 classes. We construct CLIP text prompts only for the 25 source classes, and no prompt is built for the 40 target-private unknown classes. Unknown target samples are handled following SHOT [2]: after obtaining prediction scores over the known source classes, we compute the entropy of the prediction distribution and use KMeans to separate confident known samples from high-entropy uncertain samples. The high-entropy cluster is rejected as unknown. Thus, unknown classes are detected by uncertainty-based rejection, not by assuming their class names are available.
>
> We have expanded **Sec. 4.1 (Implementation Details)** in the revised version with an additional paragraph detailing our use of [CLS].
>
>
> [1] Deep hashing network for unsupervised domain adaptation. In CVPR2017.
>
> [2] Do we really need to access the source data? source hypothesis transfer for unsupervised domain adaptation. In ICML2020.